# Cancer cell histone density links global histone acetylation, mitochondrial proteome and histone acetylase inhibitor sensitivity

Christopher Bruhn [1,3✉], Giulia Bastianello [1,2] & Marco Foiani [1,2✉]

Chromatin metabolism is frequently altered in cancer cells and facilitates cancer development. While cancer cells produce large amounts of histones, the protein component of chromatin packaging, during replication, the potential impact of histone density on cancer biology has not been studied systematically. Here, we show that altered histone density affects global histone acetylation, histone deacetylase inhibitor sensitivity and altered mitochondrial proteome composition. We present estimates of nuclear histone densities in 373 cancer cell lines, based on Cancer Cell Line Encyclopedia data, and we show that a known histone regulator, HMGB1, is linked to histone density aberrations in many cancer cell lines. We further identify an E3 ubiquitin ligase interactor, DCAF6, and a mitochondrial respiratory chain assembly factor, CHCHD4, as histone modulators. As systematic characterization of histone density aberrations in cancer cell lines, this study provides approaches and resources to investigate the impact of histone density on cancer biology.

[1] The FIRC Institute of Molecular Oncology (IFOM), Milan, Italy. [2] Università degli Studi di Milano, Milan, Italy. [3]Present address: Evotec International GmbH, Göttingen, Germany. ✉email: bruhndatascience@gmail.com; marco.foiani@ifom.eu

DNA is wrapped around histones to establish nucleosomes as basic packaging unit[1]. A nucleosome contains two copies of each core histone (H2A, H2B, H3, H4). Nucleosomes are dynamic structures which are disassembled and displaced to grant access to naked DNA during replication, transcription and DNA damage repair[2]. Nucleosome density and histone-DNA interaction influence the efficiency of these processes and are tightly regulated by chromatin assembly factors, chromatin remodellers and histone modifiers[3,4]. Nucleosome assembly requires an adequate supply of histone proteins. The dosage of histones, which is the amount of histones in relation to chomatin assembly factors and DNA, can affect nucleosome assembly and density, and thereby impact on chromatin structure[5].

The histone landscape in humans is complex[6]: while there is only one histone H4 protein, diverse variants exist for H2A, H2B and H3. These histone variants can be categorized as replication-dependent (canonical) variants, which constitute the major fraction of histones and are produced mainly during S phase, and replication-independent (non-canonical) variants, which are also expressed outside of S phase. Most replication-dependent histone variants are encoded by multiple genes, which are organized in large, coordinately regulated clusters[7]. The expression of these clusters is driven by the transcription factors NPAT, Oct-1 and HiNF-1, together with the co-activators TRRAP and Tip60[8]. The cyclin-dependent kinase 2 (Cdk2) cyclin E complex phosphorylates and activates NPAT at the onset of S phase and thereby boosts histone gene transcription[9]. The high copy number of histone genes facilitates sufficient expression to support chromatin assembly during DNA replication. Replication-dependent and -independent histone mRNAs are further regulated at the level of translation and turnover: as for most mRNAs, the stability and translation of replication-independent histone mRNAs is controlled by their 3' polyadenylated (polyA) tail[10]. In contrast, replication-dependent histone mRNAs have a 3' stem-loop structure instead of the polyA tail, which promotes efficient mRNA translation[11] and facilitates mRNA degradation upon completion of DNA replication[12]. Histone protein levels are controlled by a variety of degradation mechanisms, including protease cleavage, the autophagy-lysosome system, and the proteasome complex[13]. Chromatin structure and DNA metabolism can potently influence histone dosage. The most striking example is the high-mobility group protein B1 (HMGB1), one of the most abundant chromatin components[14], loss of which is associated with a global histone dosage reduction by 20%[5]. The DNA damage response signaling network further controls histone gene expression[15,16], mRNA stability[12] and protein turnover[17] to adjust histone dosage during DNA synthesis, replication stress response and DNA repair.

The maintenance of adequate chromatin structure is critical for cell identity[18]. Altered chromatin remodelling, chromatin modification and histone proteins frequently occur in cancers, and several of the associated chromatin changes are known to drive cancer development[19]. Histone dosage is elevated in some cancer cell lines, where it increases nucleosome occupancy[20]. However, in contrast to the above chromatin processes, the link between histone dosage and cancer biology is not well explored. Elevated histone dosage may provide a survival advantage for cancer cells by facilitating high histone supply during DNA replication, and by protecting the DNA from damaging agents[20]; however, excessively high histone levels have various documented adverse effects, including unspecific nucleic acid binding, inhibition of enzyme activities and interference with metabolic pathways[16,21]. Currently there are two major limitations in understanding whether histones dosage plays a role in cancer development. First, we are lacking a systematic characterization of histone dosage across widely used cancer models. The underlying histone quantification would need to be highly accurate because small changes in histone dosage (20% and less) are sufficient to affect nucleosome biology. Variations in DNA content and nuclear-to-cytoplasm ratio further complicate a meaningful definition of histone dosage by raising the question of the ideal quantification and normalization procedures. Second, in spite of the large number of characterized histone regulators, only little is known about potential alterations in histone regulators that have an actual impact on steady state histone levels in cancer, as in the case of HMGB1.

Most commonly used cancer cell lines have been characterized by the Cancer Cell Line Encyclopedia (CCLE) project at the level of genome, transcriptome, proteome, metabolome, genetic dependencies and drug sensitivities[22]. The CCLE datasets are a valuable resource to investigate regulatory mechanisms in silico. The large number of included cell lines provides many examples in which a gene or protein of interest shows particularly high or low expression, and thereby allows a statistical analysis of biological features that are associated with its expression[23,24].

Here, we explore the phenomenon of naturally occurring histone density variation in cancer using CCLE resources. We classify CCLE cancer cell lines by histone density, describe histone density-associated molecular signatures and drug responses and identify bona fide histone modulators.

## Results

**Histone level prediction in cancer cell lines based on proteomics data.** Aberrant histone levels have been reported for several cancer cell lines, in comparison with non-transformed cells[20]. We decided to systematically investigate the phenomenon of histone dosage in cancer cells. The Cancer Cell Line Encyclopedia (CCLE) project characterized more than 1300 cancer cell lines by RNA-Seq[22] and 375 by global proteomics[25] and hence provides the largest available resource for such systematic analysis. The datasets covered 15 core histone protein variants (H2A, H2B, H3, H4), which were encoded by 43 mRNAs (Fig. 1a). Seven of these variants were encoded by multiple mRNAs, and we assessed their total mRNA levels by integrating the expression of their individual mRNAs (see Methods, Fig. 1a and Supplementary Fig. 1a).

Histone levels could be predicted from mRNA or protein measurements. The main advantages of mRNA measurements are their lower cost, highly standardized methodology and broad availability. Their main disadvantage is the frequent use of polyA mRNA enrichment, such as in the CCLE RNA-Seq dataset[22], which causes an under-representation of replication-dependent (non-polyA) histone mRNAs[26,27]. In agreement, we found that replication-dependent histone mRNAs were less abundant than the replication-independent (polyA) histone mRNAs and other mRNAs in general (Fig. 1b), and for specific histone types (Fig. 1a, replication-dependent H3.1/H3.2 vs. -independent H3.3 mRNAs). Nonetheless, CCLE RNA-Seq data reflected the typical co-regulation pattern of replication-dependent histone mRNAs[8] (Fig. 1c and Supplementary Fig. 1b). To assess how accurately histone protein levels can be predicted by their encoding mRNAs, we calculated protein-mRNA correlation coefficients. Protein and mRNA levels correlated strongly (macroH2A2) or moderately well (macroH2A1, H2AX, H3.3) for several replication-independent histones, but poorly for all replication-dependent histones, including the ones with the highest apparent expression (H2A type 2-A: *H2AC18*, H2B type 1-J: *H2BC11*, H3.1: *H3C10*, H4: H4C14) (Fig. 1d and Supplementary Fig. 1c). An independent CCLE microarray dataset of polyA-enriched mRNAs[28] revealed a comparable lack of protein-mRNA correlation

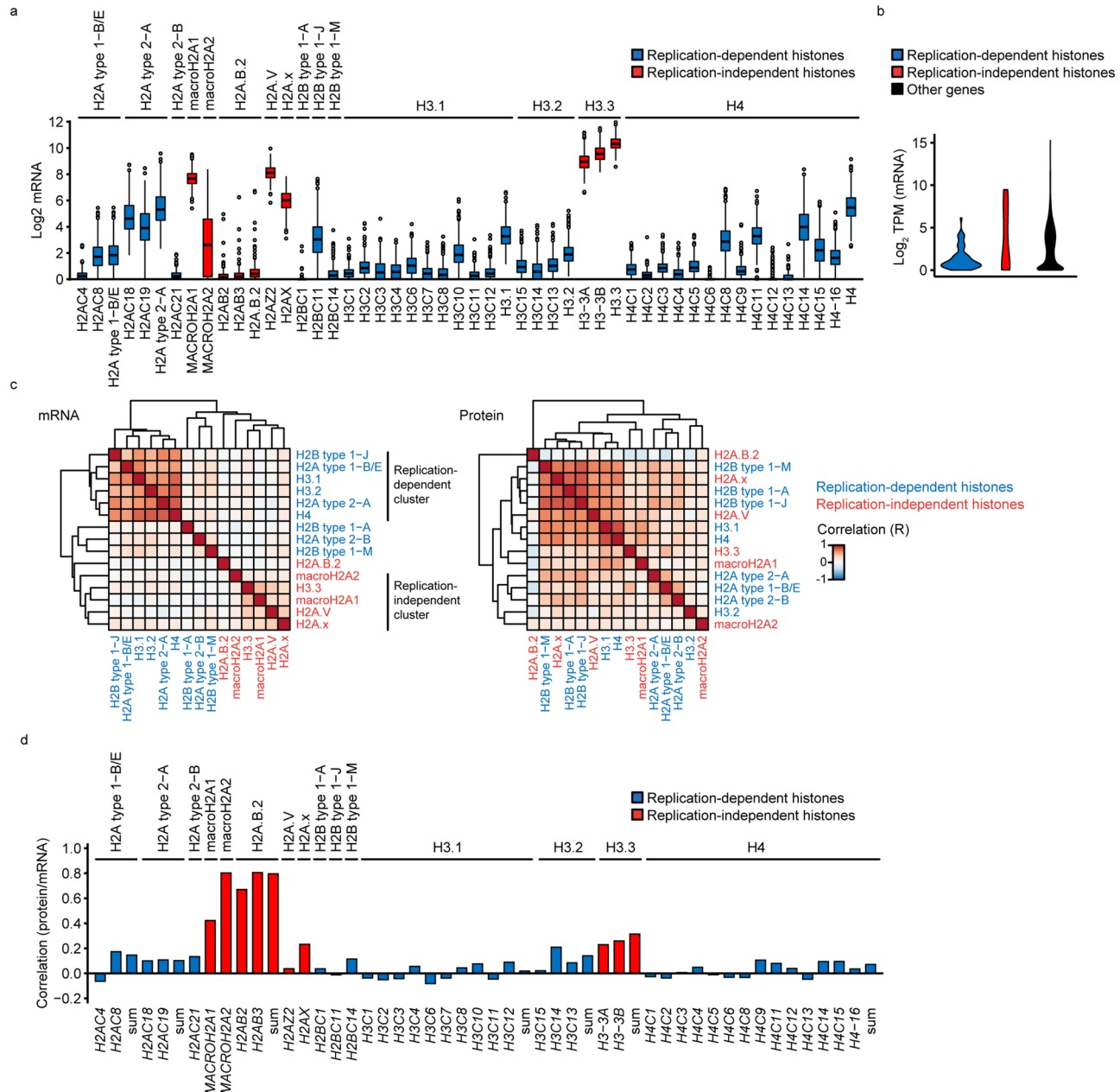

**Fig. 1 Histone mRNA and protein expression in CCLE datasets. a, b** Histone mRNA levels in cancer cell lines. **a** The boxplots show the histone mRNA read distribution across cancer cell lines from the CCLE expression (RNA-Seq) dataset for which proteome data are available. Histone gene names are specified by x axis labels. Histone genes are grouped by the encoded histone variant as indicated above the boxplots. The x label sum indicates that the boxplot represents the read sum of all detected histone transcripts encoding the respective variant. **b** Mean read counts of replication-dependent and -independent histone genes were calculated for each CCLE cell line. The violin plots display their distribution across cell lines. The colors represent the histone classification by replication dependence. **c** Correlation of histone mRNAs and proteins across cancer cell lines. The heatmaps represent Pearson correlation coefficients of estimated histone mRNA variant levels and histone protein variant levels from the CCLE RNA-Seq expression and proteomics datasets, organized by hierarchical clustering. Font colors indicate histone classification by replication dependence. **d** Correlation of histone proteins vs. mRNAs across cancer cell lines. The barplots represent Pearson correlation coefficients of histone protein variants vs. the encoding mRNAs. Axis labels and color coding are as in (**a**). Cell lines covered by both CCLE RNA-Seq and proteomics datasets (N = 372) are represented. Only histone genes are represented of which the encoded protein variant is covered by the proteomics datasets. Data in (**c, d**) were lineage-centered. TPM transcripts per million.

(Supplementary Fig. 1d). We extended our analysis to datasets from The Cancer Genome Atlas (TCGA) Network and the Clinical Proteomic Tumor Analysis Consortium, which provide patient-matched transcriptomics and proteomics datasets for various cancer types. These include breast cancer[29] and ovarian cancer[30] microarray datasets in which mRNA enrichment has been conducted by ribosomal RNA subtraction instead of polyA enrichment. We found a similarly poor correlation for

replication-dependent histone proteins and mRNAs (Supplementary Fig. 1d). We observed the same lack of correlation for cancer studies based on RNA-Seq with polyA mRNA enrichment, including colon adenocarcinoma[31], lung squamous cell carcinoma[32,33], glioblastoma multiforme[34,35], head and neck squamous cell carcinoma[36,37] and uterine corpus endometrial carcinoma[38,39] (Supplementary Fig. 1d). In summary, our analysis suggests that histone mRNA measurements do not

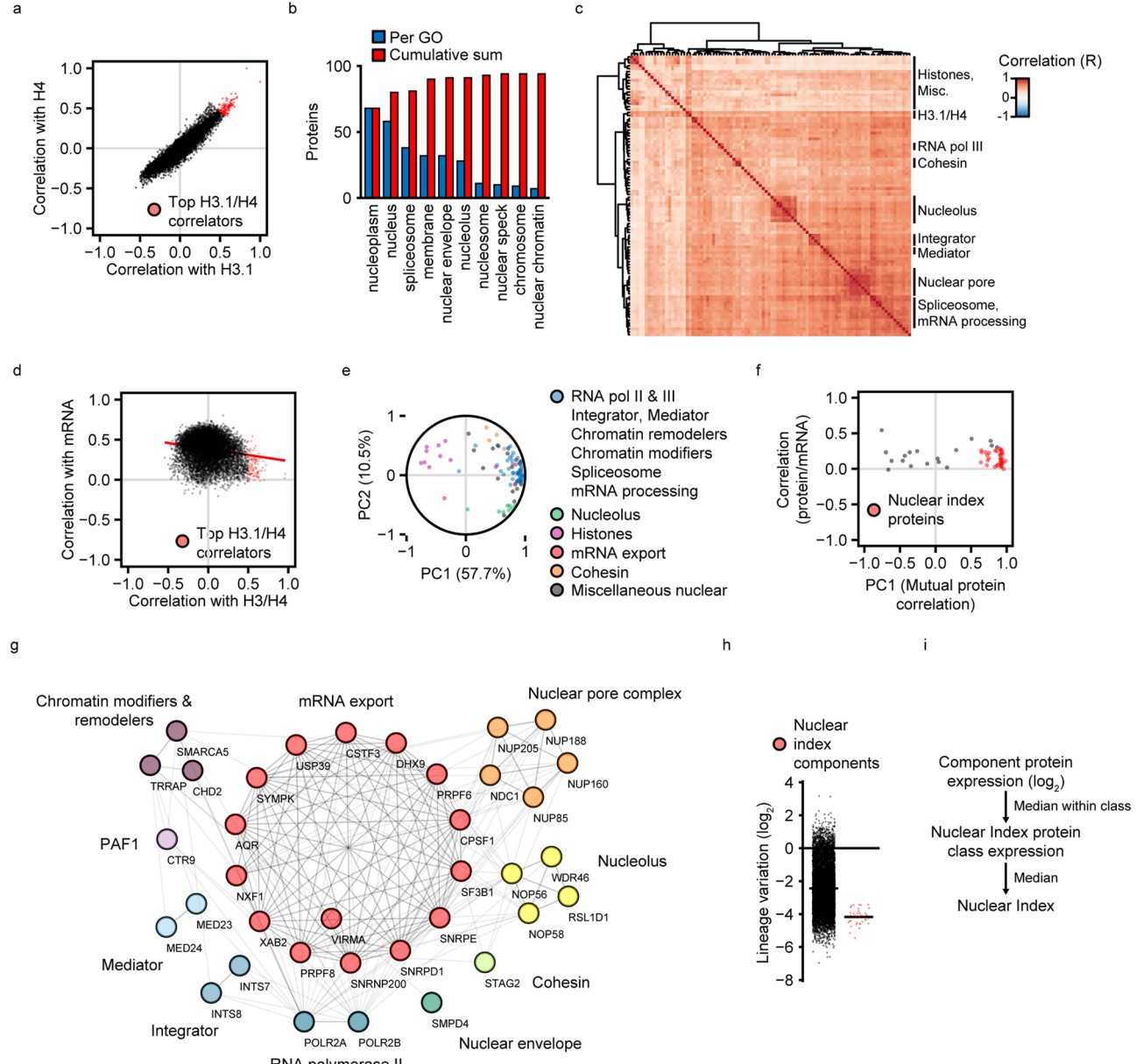

reflect histone protein levels across cancer cell lines or patient samples.

**A nuclear index as reference for histone density estimation.** We explored the CCLE proteomics dataset to investigate the phenomenon of histone density. CCLE proteomics data reflect the level of individual proteins in relation to the total proteome. Histone levels are therefore influenced by the relative contribution of the nuclear proteome to the total proteome. An estimate of nuclear histone density should therefore relate histone levels to the nuclear proteome. Nuclear reference proteins should appear as co-expressed with histones and each other, they should represent diverse nuclear structures, and their expression should not be cell line-specific.

To identify suitable nuclear reference proteins, we first scored all proteins for co-expression with histones. We used H4 as model histone, because it exists as single variant and hence does not require the consideration of isoform complexity, and H3.1, based on its co-expression with H4 (see Fig. 1c right panel). Gene ontology (GO) cellular component analysis revealed that the top

H3.1/H4 co-expressed proteins (Fig. 2a; R > 0.5) were almost exclusively nuclear (Fig. 2b). Most of these proteins were highly co-expressed among each other (Fig. 2c, enlarged image in Supplementary Fig. 2) and included subunits of diverse constitutive nuclear complexes (Fig. 2c and Supplementary Fig. 2). Thus, based on their expression and functional diversity, these nuclear proteins may serve as nuclear reference for histone expression. The level of these proteins correlated poorly with their mRNAs (Fig. 2d), implying robustness against changes in gene expression. We created a robust subgroup of highly co-expressed nuclear proteins with particularly low protein/mRNA correlation, using simplification of protein co-expression by principal component analysis (PCA) (Fig. 2e, f). The resulting proteins were divided into 10 categories: mRNA export factors, the nuclear pore complex, nucleolar proteins, chromatin modifiers/remodellers, RNA pol II, Mediator, Integrator and PAF1 complexes, cohesin, and the nuclear envelope (Fig. 2g). All selected proteins had a low inter-lineage variation (Fig. 2h). We calculated a representative nuclear protein expression value, the nuclear index, from the nuclear reference proteins (Fig. 2i).

**Fig. 2 Definition of a nuclear reference proteome. a** Protein correlations with histones H3.1 and H4 across cancer cell lines. The scatter plot shows the Pearson correlation coefficients (R) of proteins vs. H3.1 (*x* axis) and H4 (y axis). Each dot represents a protein covered by the CCLE proteome dataset. Proteins that correlate strongly with both histones (R > 0.5) are colored in red. **b** Gene ontology cellular compartment analysis of top histone correlating proteins from (**a**). Significantly over-represented cellular components were sorted by the number of top histone correlators localized to the respective component. The blue bars show the number of top histone correlators covered by component. The red bars show the cumulative sum of the top histone correlators covered. **c** Mutual correlations between top histone correlating proteins from (**a**). The heatmap represents Pearson correlation coefficients of protein levels, organized by hierarchical clustering. An enlarged version with protein labels is available in Supplementary Fig. 2. **d** Correlation of protein vs. mRNA levels across cancer cell lines. The *y* axis indicates the Pearson correlation coefficient of each protein in the CCLE proteomics dataset vs. its encoding mRNA. The *x* axis indicates the mean Pearson correlation coefficient of each protein vs. histones H3.1 and H4. Top histone correlating proteins from (**a**) are colored in red. The red line represents the quantile regression. **e** Principal component analysis of top histone correlator co-expression. Pearson correlation coefficients from **c** were simplified by PCA. The PCA correlation plot shows the first two principal component correlations. Each dot represents one top histone correlating protein. Dot positions reflect the clustering behavior in **c**. Colors indicate the protein classification by complex and function. **f** Selection of proteins for the representative nuclear index. The y axis indicates the protein-mRNA correlation (same as the y axis in **d**). The x axis indicates the first principal component of top histone correlator co-expression (same as the x axis in **e**). Proteins with low protein-mRNA correlation (R < 0.3) and a similar co-expression spectrum (PC1 > 0.5), marked in red, were included in the nuclear index. **g** Components of the representative nuclear index. Proteins selected in (**f**) were clustered by STRING and manually grouped into 10 categories with Cytoscape. **h** Lineage variation of nuclear index components. Mean levels of each protein were calculated by cell lineage. The relative variation between lineages is represented as the standard deviation of lineage means divided by the mean of lineage means. **i** Scheme for the calculation of the nuclear index. To ensure a balanced contribution of all nuclear index protein categories, the nuclear index is calculated in two steps: first, individual protein expressions are converted to representative category values. Second, the representative category values are integrated. This method eliminates the effect of potential large expression changes of individual proteins by using median values across diverse protein categories. Cell lines covered by both CCLE RNA-Seq and proteomics datasets (N = 373) were used. Protein expression data were lineage-centered.

---

**Cell line classification by nuclear index-corrected histone densities**. We estimated nuclear histone densities by correcting histone levels for the nuclear index (Fig. 3a). Nuclear index correction strongly reduced nearly all positive and negative correlations of histones H3.1 and H4 with other proteins (Fig. 3b and Supplementary Fig. 3a). Importantly, co-expression of H3.1 and H4 with other histone proteins was unaffected by nuclear index correction (Fig. 3b and Supplementary Fig. 3a), which is expected from histone complex stoichiometry. This implies that nuclear index correction enriches for meaningful co-expression by eliminating systematic differences in nuclear vs. total protein content.

We asked if the lineage of a cancer cell line had an effect on its histone levels and the nuclear proteome. Few cancer lineages had extreme average nuclear index values, associated with corresponding shifts in the mean expression of H3.1 and H4 (Fig. 3c). Nuclear index correction reduced these systematic histone level shifts across lineages (Fig. 3c). However, the combination of nuclear index correction and lineage centering resulted in the strongest reduction of histone density variation across cancer cell lines (Fig. 3d), emphasizing the use of both methods when comparing histone density across lineages.

Histone level changes of less than 20% cause detectable differences in nucleosome density and gene expression[20]. We therefore classified CCLE cell lines by histone density using a 20% change of nuclear index-corrected H3.1 and H4 levels. Due to the lack of data on lineage-matched non-cancer counterparts, we defined high or low histone density using the lineage median of cancer cell lines. This approach classified 46 (12%) and 31 (8%) cell lines from various lineages as histone-high and -low, respectively (Fig. 3e, f, Supplementary Data 1), resulting in less cell lines with extreme histone dosage than an equivalent classification without nuclear index correction (Supplementary Fig. 3b). The nuclear index and histone gene copy numbers were comparable between histone-high and -low groups (Supplementary Fig. 3c, d), implying that nuclear protein content and histone gene dosage did not drive this classification. As validation, we quantified histone expression in histone-high and -low cell lines of three lineages (breast, skin, central nervous system) (Supplementary Fig. 4a) by Western blotting, using two nuclear index proteins, DHX9 and RNA polymerase II subunit RPB1 (POLR2A), as reference. The histone levels measured by Western blot were in good agreement with the predicted H3 and H4 levels:

all tested histone-high cell lines had average H3/H4 levels above the lineage mean, whereas all tested histone-low cell lines had average H3/H4 levels below the lineage mean (Fig. 3g, h). Classification of the same cell lines by histone expression values without nuclear index correction failed to classify 3 out of 5 cell lines as histone-high (WM2664, IPC298, MEWO) and 2 out of 5 cell lines as histone low (MDAMB436, IGR1) (Supplementary Fig. 4b). Since nuclear index calculation is not based on DNA content data, nuclear index-corrected histone densities do not directly reflect histone:DNA ratios, which are relevant for most biological effects of histones. To address how nuclear index-corrected histone densities relate to histone:DNA ratios, we performed quantitative flow cytometry analysis of DNA content in the histone-high and -low cell lines using the DNA binding dye 4′,6-diamidino-2-phenylindole (DAPI) (Supplementary Fig. 4c). There were no systematic differences of DNA content between histone-high and -low cell lines (Supplementary Fig. 4d). Moreover, DAPI-normalized average H3/H4 levels of all tested histone-high and -low cell lines were above and below the lineage mean, respectively (Supplementary Fig. 4e); hence nuclear index-corrected histone density classification was in good agreement with histone-DNA ratio across cell lines.

**The histone density-associated proteome and transcriptome**. We next investigated if histone density was linked to specific proteome signatures and analyzed the CCLE proteome dataset for differentially expressed proteins in histone-high vs. -low cells. We identified 123 and 263 proteins with increased or reduced expression in histone-high cells, respectively (FDR < 0.1, fold-change >1.3) (Fig. 4a, Supplementary Data 2, Supplementary Fig. 5a). Based on their co-expression, these differentially expressed proteins organized into 4 clusters associated with high (up in histone-high vs. -low) and low (down in histone-high vs. -low) histone density (Fig. 4b, enlarged in Supplementary Fig. 5b). Histone-high cells had a high expression of other histones apart from H3.1 and H4, regulators of chromatin structure (CBX5, CHTOP, HMGN2, MBD1, MBD2, SMARCA5), chromosome conformation (CDCA2, CDCA5, RMI2), nuclear architecture (TMPO), the CENPA nucleosome-associated complex (CENPC, CENPN, CENPQ, CENPU), transcription regulators (ELF2, ARHGAP11A, SS18L1), and mitotic spindle

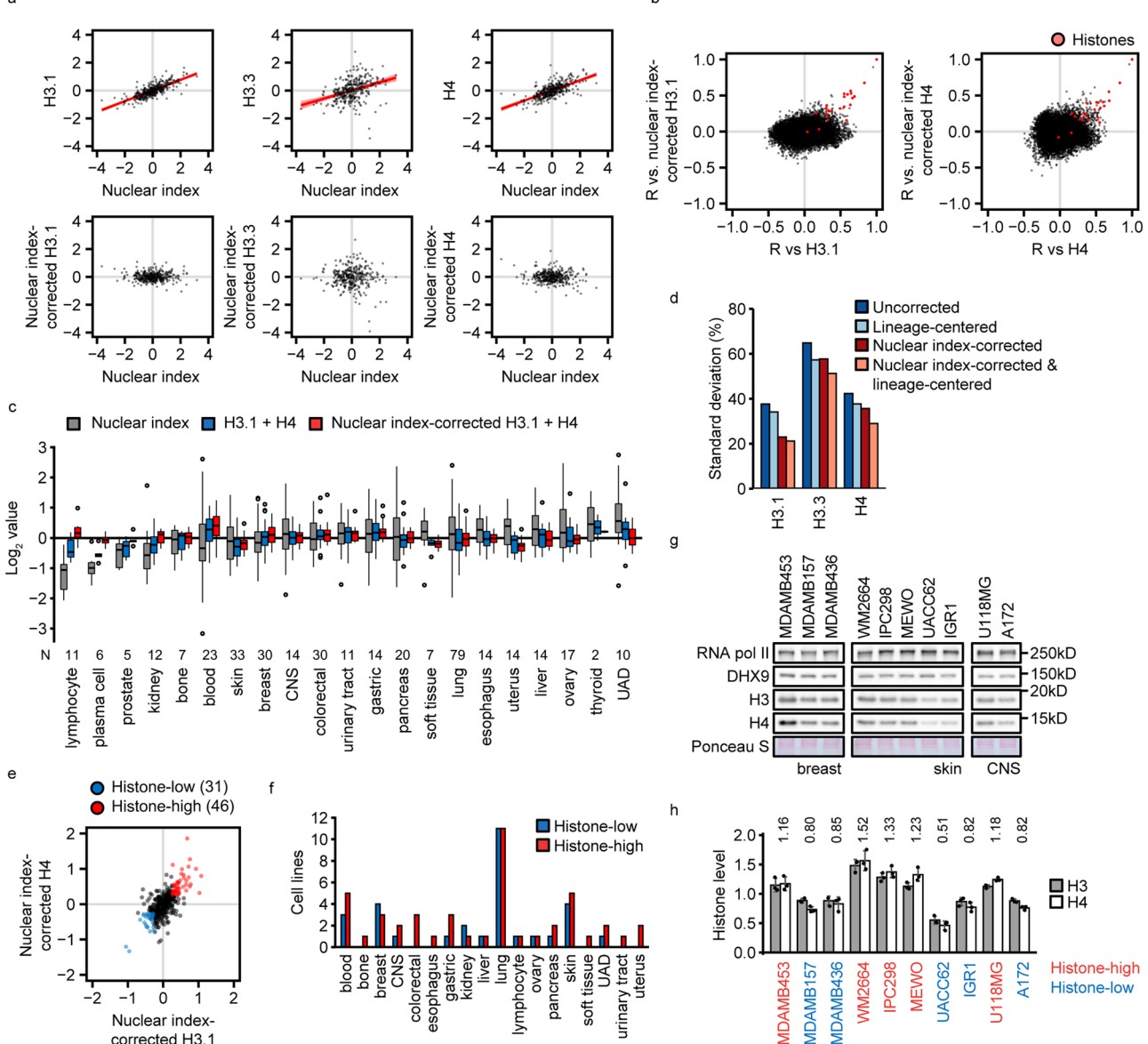

organizers (KIF22, NUMA1, SAPCD2) (cluster A). High mobility group proteins, including HMGB1, were positively associated with histone dosage (cluster B). Remarkably, knock-down of HMGB1 reduces histone H3 levels[5], implying a potential role in the variation of histone density in cancer. Other proteins associated with the histone-high state were involved in cell-environment interaction, signaling and intracellular trafficking (cluster C) and various aspects of nuclear biology (cluster D: histone demethylase KDM2A, PARP1, cohesin subunit STAG1, LINC component SYNE2). Histone-high cells had a low expression of plasma membrane, ECM and cytoskeletal proteins (cluster E: Moesin, Laminins, Collagens), various mitochondrial and rRNA synthesis proteins (cluster F), trafficking and signaling factors (cluster G: mTORC1 subunit AKT1S1, NFkappaB subunit RELB, autophagy factor ATG13), and protein degradation enzymes (cluster E: histone-cleaving protease Cathepsin L (CTSL)[40], cluster G: proteasome subunits PSMA4 and PSMC5). Gene ontology analysis confirmed that nuclear chromatin terms were associated with histone-high state, whereas mitochondrial terms were associated with histone-low state (Fig. 4c). Western blot analysis of two mitochondrial proteins, SCO1 and ALAS1, confirmed their predicted differential expression (Fig. 4d, e). In

contrast, the mitochondrial protein COX4, which was not predicted to be differentially expressed, did not correlate with histone levels (Fig. 4d, e). This implies that histone dosage is linked to specific mitochondrial proteome signatures but not to a global alteration of mitochondrial proteins. Consistently, confocal imaging of mitochondria did not reveal obvious systematic differences in mitochondria content or network structure in histone-high and -low cells (Supplementary Figs. 6 and 7). In summary, histone density associates with specific proteome signatures across cancer cell lines, and is linked to the expression of chromatin components, mitochondrial proteins, and known histone regulators (HMGB1, CTSL).

We asked to which extent gene expression contributed to the histone density-associated proteome and tested the respective mRNAs for differential expression in histone-high vs. -low cell lines (FDR < 0.1). The mRNAs of 96 out of 123 up-regulated proteins and 115 out of 263 down-regulated proteins were consistently altered in histone-high vs. -low cells, including the histone regulators *HMGB1* and *CTSL* (Fig. 4f, Supplementary Data 3). Hence, differential mRNA levels potentially account for the majority of up-regulated and nearly half of the down-regulated proteins. To identify potential transcriptional

**Fig. 3 Cancer cell line classification by histone density. a** Nuclear index correction of histone levels. Histone levels were plotted against the nuclear index, and the systematic effect of the nuclear index on histone levels across cancer cell lines was determined by quantile regression (red lne). Histone levels were then corrected for the nuclear index contribution. Each dot represents a cell line. **b** Effect of nuclear index correction on protein correlations with histones H3.1 and H4 across cancer cell lines. The scatter plots show the Pearson correlation coefficients (R) of proteins vs. H3.1 and H4 with (y axis) and without (x axis) nuclear index correction of H3.1 and H4 levels. Each dot represents a protein. Histones are colored in red. **c** Influence of cell lineage on nuclear index and histone levels. The barplots represent nuclear index and mean levels of H3.1 and H4 with and without nuclear index correction by lineage. Data are represented as $\log_2$ mean values over cell lines ± standard deviation. **d** Effect of nuclear index correction and lineage centering on histone levels. Histone levels were subjected to lineage centering, nuclear index correction or both. The standard deviation divided by the mean level is shown as measure of relative variation across cell lines. **e** Classification of cancer cell lines by histone density. The scatter plot shows nuclear index-corrected histone H3.1 and H4 levels in 373 cell lines. Each dot is a cell line. Red and blue colors indicate cell lines with at least 20% increase or decrease, respectively, of both histones H3.1 and H4. **f** The number of histone-high and -low cell lines from (**e**) by cell lineage. Note that lung cancers are over-represented in the CCLE proteomics dataset, and lung cancers do likely not have a particular tendency for extreme histone dosage. **g, h** Validation of histone density classification. **g** Protein lysates were made from asynchronous cultures of the indicated cell lines from 3 different lineages. Protein levels were analyzed by Western blot. **h** The barplots represent histone levels normalized to the mean of RNA polymerase II and DHX9. Data are represented as mean of 3 independent parallel replicate cultures ($N = 3$) ± standard deviation, with the lineage mean set to 1. The numbers above the bars represent the mean of H3 and H4 for the respective cell lines. The font color indicates the cell line classification from (**e**) by histone density. Significances were calculated by lineage with one-way ANOVA (breast: $p_{\text{ANOVA, H3}} = 0.012$, $p_{\text{ANOVA, H4}} = 4.0 \times 10^{-3}$; skin: $p_{\text{ANOVA, H3}} = 3.0 \times 10^{-7}$, $p_{\text{ANOVA, H4}} = 1.4 \times 10^{-6}$) with post hoc Tukey HSD test (breast: $p_{\text{MDAMB453 vs. MDAMB157, H3}} = 7.7 \times 10^{-4}$, $p_{\text{MDAMB453 vs. MDAMB157, H4}} = 0.020$, $p_{\text{MDAMB453 vs. MDAMB436, H3}} = 1.6 \times 10^{-3}$, $p_{\text{MDAMB453 vs. MDAMB436, H4}} = 0.019$; skin: $p_{\text{WM2664 vs. UACC62, H3}} = 1.8 \times 10^{-7}$, $p_{\text{WM2664 vs. UACC62, H4}} = 2.7 \times 10^{-7}$, $p_{\text{WM2664 vs. IGR1, H3}} = 5.6 \times 10^{-6}$, $p_{\text{WM2664 vs. IGR1, H4}} = 1.3 \times 10^{-5}$, $p_{\text{IPC298 vs. UACC62, H3}} = 1.3 \times 10^{-6}$, $p_{\text{IPC298 vs. UACC62, H4}} = 2.6 \times 10^{-6}$, $p_{\text{IPC298 vs. IGR1, H3}} = 1.0 \times 10^{-4}$, $p_{\text{IPC298 vs. IGR1, H4}} = 3.9 \times 10^{-4}$, $p_{\text{MEWO vs. UACC62, H3}} = 4.3 \times 10^{-6}$, $p_{\text{MEWO vs. UACC62, H4}} = 2.2 \times 10^{-5}$, $p_{\text{MEWO vs. IGR1, H3}} = 6.3 \times 10^{-4}$, $p_{\text{MEWO vs. IGR1, H4}} = 0.011$) or two-sided, unpaired Student's t test (CNS: $p_{\text{U118MG vs. A172, H3}} = 5.6 \times 10^{-4}$, $p_{\text{U118MG vs. A172, H4}} = 5.1 \times 10^{-5}$). Nuclear index analysis (**a–f**) was applied to all CCLE proteome cancer cell lines ($N = 373$). Protein expression data were lineage-centered unless otherwise indicated. CNS central nervous system, UAD upper aerodigestive tract.

programs, we applied regulatory target gene set enrichment analysis. mRNAs with high expression in histone-high cells were enriched for targets of E2F, Snail (SNAI1) and YY1 transcription factors, covering in total 31 out of 96 mRNAs. mRNAs with low expression in histone-high cells were enriched for targets of NFκB, STAT5B, SRF, BACH1 and AP1, but only 18 out of 115 altered mRNAs were controlled by these transcription factors (Fig. 4g, Supplementary Data 4–5). Hence, our data implies a potential involvement of several transcription factors in shaping the histone-high proteome signature.

**Histone density predicts global acetylation state and drug resistance.** The histone density-associated proteome may be used as molecular signature to predict histone density. We developed a logistic model for histone density prediction based on PCA-simplified histone density-associated proteins. The model efficiently distinguished histone-high from -low cell lines (Fig. 5a). 10 cell lines of particularly high histone density (>30% increase) were clearly distinguished with high confidence (>99.9%) from all cell lines with normal and low histone density (Fig. 5a–c). We argued that such cell lines are most likely to display other characteristics associated with high histone density and analyzed CCLE datasets (global chromatin profiling[22], Sanger drug sensitivity[41,42], Achilles CRISPR/Cas9 gene dependency[43], DEMETER2 siRNA gene dependency[44] and metabolomics[23]) for features that were significantly altered in these cell lines, compared to cell lines with normal or low histone density. High histone dosage was associated with elevated acetylation of histone H3 on lysines 14, 18, 23 and 27 (Fig. 5d). Histone-high cells were also sensitive to two histone deacetylase (HDAC) inhibitors, trichostatin-A and panobinostat (Fig. 5e), implying that the already elevated global histone acetylation sensitizes towards a further increase in acetylation. Notably, high histone density was linked to several other drug sensitivities and resistance towards four differernt MEK1/2 inhibitors (Fig. 5e). Moreover, histone-high cells had an increased resistance to the targeting of *ZC3H13*, an N6-methyladenosine methylation factor for mRNAs, by siRNA and CRISPR/Cas9 technologies (Fig. 5f). Metabolic alterations linked to the histone-high state were entirely accounted for by lineage effects (Supplementary Fig. 8). In summary, our analysis shows that cancer cell lines with highly elevated histone dosage show high levels of histone acetylation and altered responses to several cancer drugs, of which HDAC inhibitors directly affect the histone modifications.

**Identification of histone dosage modulators.** Histone density-associated proteins may be co-regulated with histones, histone-regulated, or histone modulators, such as HMGB1, which could drive or buffer abnormal histone densities. We conducted a small-scale siRNA high-content imaging screen of cell cycle-corrected H3 levels (Fig. 6a–c)[45] in the triple-negative breast cancer cell line MDA-MB-231, which is characterized by normal histone density, rendering a pre-existing alteration of histone density modulators less likely (Fig. 6a). We selected candidates strongly associated with histone density by ranking all proteins of the histone density-associated proteome by their ability to predict histone dosage (Supplementary Figs. 9a–f, Supplementary Data 6–7). We identified 4 candidate siRNAs, of which 2 decreased (si*CHCHD4*, si*DCAF6*) and 2 increased (si*TMPO*, si*PSMC5*) H3 (Fig. 6d). Of these, si*CHCHD4* and si*TMPO* had no major impact on cell viabiliy and cell cycle distribution, whereas si*DCAF6* and si*PSMC5* reduced cell viability and caused a cell cycle shift towards G2/M phase (Supplementary Fig. 10a, b). The ratio of histone H3 intensity vs. DAPI was not influenced by cell loss (Supplementary Fig. 10c). We conducted two validation screens for the 4 candidates, using the same polyclonal H3 antibody or a co-staining of two monoclonal anti-H3 and anti-H4 antibodies (Fig. 6e, f). Both H3 antibodies validated the primary screen, whereas the H4 antibody validated si*CHCHD4*, si*DCAF6* and si*PSMC5*, but not si*TMPO* (Fig. 6f and Supplementary Fig. 10d). The effects on histone signal were cell cycle-independent in si*CHCHD4*, si*TMPO* and si*PSMC5* conditions, whereas si*DCAF6* depleted histone signal more profoundly in S and G2/M phases (Supplementary Fig. 10e). Notably, all candidate siRNAs reduced incorporation of the thymidine analog EdU into DNA in S phase (Supplementary Fig. 10f), reflecting a reduction of replication rate. si*DCAF6* also caused an expansion of the nucleus (Supplementary Fig. 10g). We performed candidate validation by Western blotting. si*CHCHD4* and si*DCAF6* reduced histones H3 and H4 (Fig. 6g), whereas si*TMPO* and si*PSMC5* did not elevate histone levels (Supplementary Fig. 10h). Thus, the immunocytochemical quantification of si*TMPO*- and si*PSMC5*-

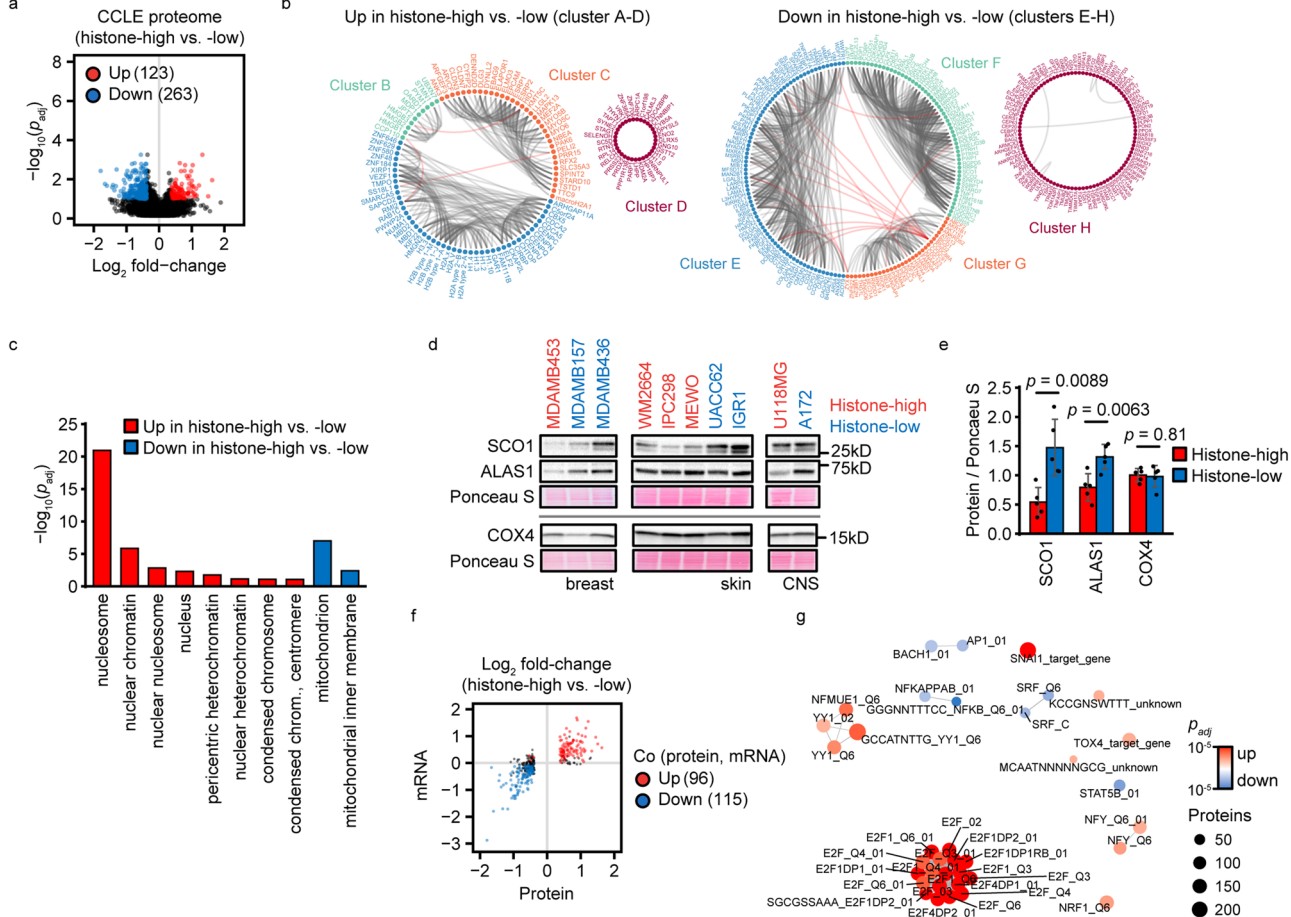

**Fig. 4 Histone density-associated molecular signatures. a, b** Significantly different proteins in histone-high ($N = 37$) vs. -low ($N = 31$) cells. **a** Proteins were tested for significantly different levels in histone-high vs. low cells using the R package maanova (FDR < 0.1, fold-change >1.3). Significant hits are shown in the Volcano plot by color coding. **b** Proteins with significantly higher or lower expression in histone-high cells were clustered by co-expression across cancer cell lines, and clusters were visualized by edge bundling. Grey and red edges represent co-expression within and between clusters, respectively. A magnification of the panel is provided in Supplementary Fig. 5b. **c** GO slim cellular component analysis of differentially expressed proteins from (**a**). **d, e** Mitochondrial protein analysis by Western blot (**d**) and quantification based on lineage-centered protein levels normalized to Ponceau S staining of total proteins (**e**). Significance was analyzed with two-tailed Student's *t*-test comparing the expression of the indicated proteins in histone-high ($N = 5$) vs. -low ($N = 5$) cell lines ± standard deviation. **f** Significantly different mRNAs in histone-high ($N = 37$) vs. -low ($N = 31$) cells. mRNAs encoding candidate proteins from (**a**) were tested for significantly different levels in histone-high vs. -low cells using the R package maanova (FDR < 0.1). Significant hits with co-regulation of protein and mRNA are shown in the scatter plot by color coding. **g** Transcription factor target gene set enrichment analysis of histone density-associated mRNAs. mRNA fold-changes in histone-high vs. -low cells (**f**) were subjected to gene set enrichment analysis using the transcription factor target signature collection. Significant signatures (FDR < 0.1) were clustered and visualized by emapplot (R package clusterprofiler). The node color indicates the direction of signature regulation.

treated cells do not reflect differences in histone protein levels, but are likely caused by improved epitope accessibility of H3 and H3/H4, respectively. We confirmed that the siRNAs used in the screen and validation experiments reduced expression of CHCHD4 and DCAF6 (Fig. 6h–j). Together, we identify CHCHD4 and DCAF6 as bona fide modulators of histone density.

## Discussion

We developed a method to estimate histone density based on a CCLE proteome dataset and used it to establish a histone density classification of cancer cell lines (Supplementary Data 1), which can serve as resource to investigate the biological effects of histone density in lineage-matched cell models. We explored proteome, transcriptome, drug resistance and epigenetic modification patterns associated with histone density across cancer cell lines, revealing links between histone density, mitochondrial proteome

composition, histone hyper-acetylation and altered cancer drug sensitivities (Fig. 7). We also identified known histone modulators as histone density-associated proteins (HMGB1, CTSL) and report two potential histone modulators.

High histone density is associated with the elevated expression of various chromatin components, which suggests a coordinated or adaptive regulation of histones along with other key chromatin components. One of these chromatin proteins is HMGB1, which is required to maintain high histone levels in cancer cells[5]. Our study therefore implies that differences in HMGB1 expression contribute to natural histone density variation across cancer cell lines. The protease cathepsin L and two proteasome subunits (PSMA4, PSMC5) were inversely associated with histone dosage. Cathepsin L catalyzes N-terminal histone H3 cleavage in embryonic stem cells[40] and in intestinal villi[46] during differentiation. Histone H3 cleavage by cathepsin L has also been shown to mediate oncogene-induced and replicative senescence[47]. The inverse correlation between cathepsin L levels and histone density raises the possibility

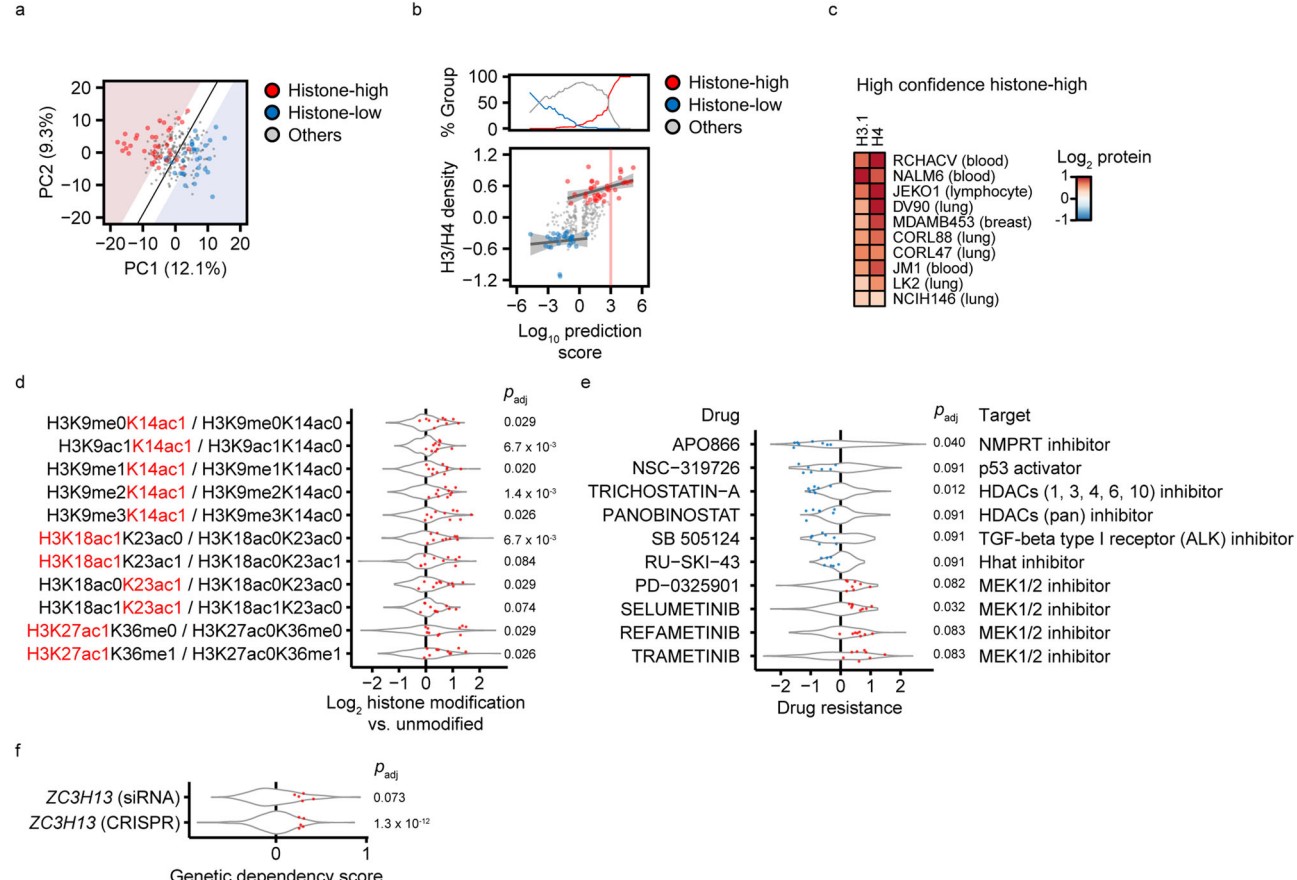

**Fig. 5 Characterization of the histone-high state by comparative OMICs. a** Cell line classification by histone density. PCA was applied to the expression data of significant proteins from Fig. 4a. The first two principal components are plotted and each dot is one cell line. A logistic model was trained for the classification of histone-high (red, $N = 37$) vs. -low (blue, $N = 31$) cell lines. The red and blue areas mark a 95% classification confidence for histone-high and -low cell lines, respectively. Cell lines with normal histone content are shown in grey ($N = 209$). **b** Contribution of histone-high ($N = 37$), -low ($N = 31$) and other ($N = 209$) cell lines at a given prediction score from **a**, and histone density vs. the cell line classification confidence from **a**, where each dot is one cell line. Quantile regressions were separately performed for histone-high and -low cells. The $\log_{10}$ prediction score reflects the likelihood that a cell line is correctly classified as histone-high (positive range) or -low (negative range). Note that a strong prediction score for the histone-high state (>3), indicated by the red separator bar, is specific for cell lines with particularly high histone dosage and separates them from histone-low and other cells. **c** Summary of high confidence histone-high cells ($N = 10$) from (**b**) with representation of H3.1 and H4 levels as heatmap. **d–f** Characteristics of high histone density in CCLE datasets. Datasets from global chromatin profiling (**d**), drug sensitivity (**e**) and gene dependency (**f**) were analyzed for significant differences between high confidence histone-high cell lines (**c**, $N = 10$) vs. cell lines with normal or low histone density ($N = 240$). All data were lineage-centered. The distribution of values in cell lines with normal or low histone density is indicated by violin plots. The distribution of values in high confidence histone-high cell lines is overlaid as blue dots (significant decrease) or red dots (significant increase). Significance between the two groups was analyzed with two-tailed, unpaired Student's $t$-test with Benjamini-Hochberg correction ($p_{adj} < 0.1$, fold-change >1.2). All significant histone modifications, drugs and gene dependencies are shown.

that its expression may influence histone turnover in cancer cells through cleavage. The proteasome mediates histone degradation in vivo and thereby determines the rate of nucleosome turnover[48,49]. Notably, low protein level of the proteasome component PSMC5 is the proteome-wide strongest predictor of high histone density (Supplementary Fig. 9a, b). Proteasome inhibitors are already in use for the treatment of selected cancers[50]. It would therefore be interesting to investigate if a reduction of proteasome activity causes elevated histone density.

We identify an inverse correlation between histone density and a large group of mitochondrial proteins (Fig. 4b, c). While altered histone density is not associated with obvious changes in global mitochondrial network structure or content (Supplementary Figs. 6 and 7), it would be interesting to investigate a potential impact on the various metabolic mitochondrial functions in the future. Notably, we demonstrate that siRNA targeting the mitochondrial protein CHCHD4, a crucial factor in respiratory

chain assembly[51], reduces histone levels in the MDA-MB-231 breast cancer cell line (Fig. 6f, g). Since CHCHD4 inversely correlates with histone density (Fig. 4b), we can exclude that its expression is causative for the differences in histone levels between cancer cell lines. Instead, our data would support a model in which a subset of genes encoding mitochondrial proteins is repressed by excessive histones, which induces a compensatory feedback loop to lower histone levels, as in the case of siRNA-mediated CHCHD4 depletion. Such feedback loop is expected to control the expression of mitochondrial protein-encoding genes that are located in the nuclear chromatin, but not of such genes that are located in the histone-free mitochondrial DNA. Consistently, all mitochondrial proteins with low expression in histone-high cells were encoded in the nucleus (Supplementary Fig. 5b). This regulatory model is in agreement with a report linking high histone levels to repression of mitochondrial functions[52] (Fig. 7). Mechanistically, high histone levels could

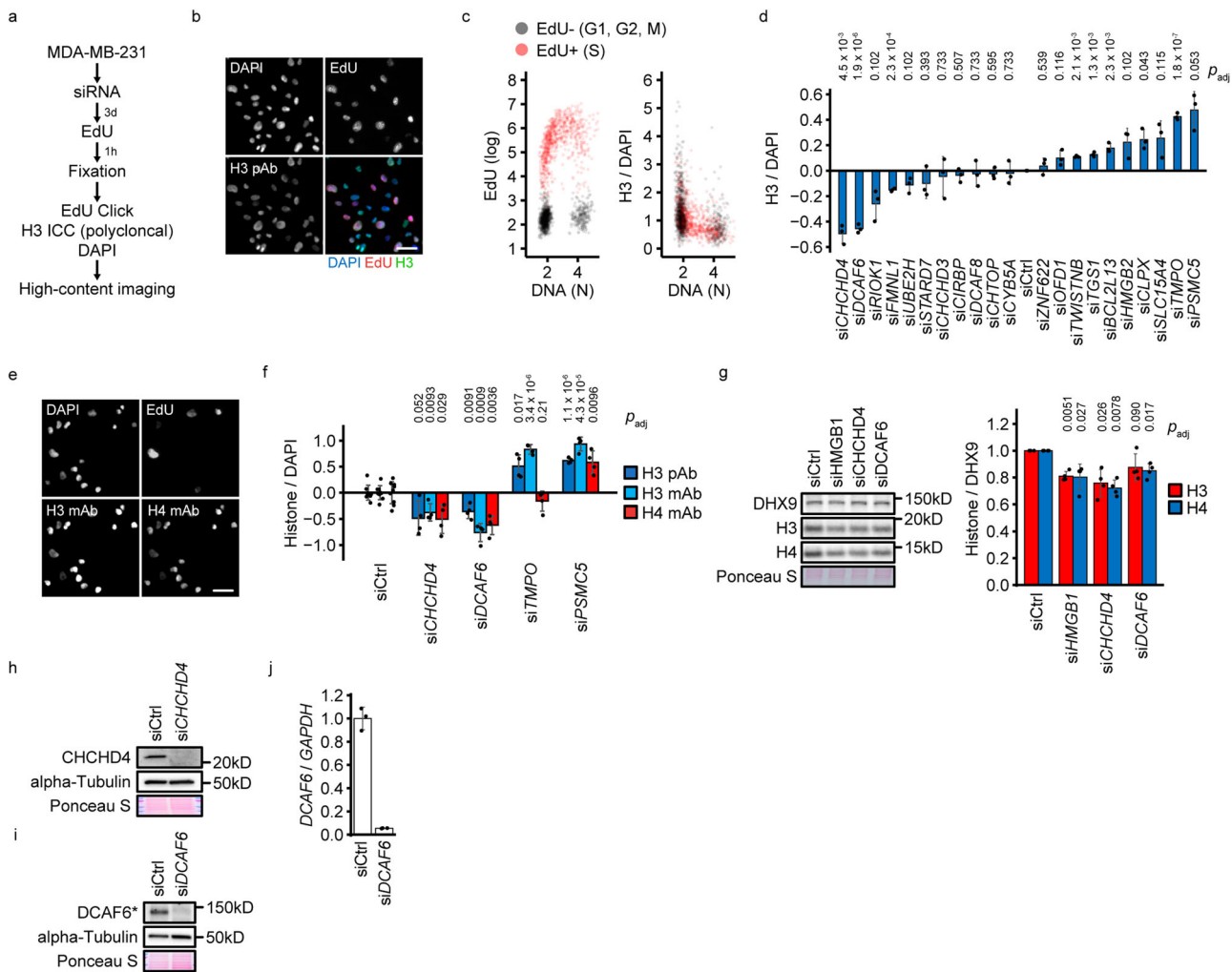

**Fig. 6 Screen for histone level modulators. a** Scheme of the siRNA screen for histone density modulators. **b, c** High-content imaging analysis of histone H3. MDA-MB-231 cells were transfected with the scrambled control siRNA and processed as in (**a**). Representative microscopy images are shown in (**b**). Examples for image-based cell cycle phase analysis and histone quantification are shown in (**c**). **d** Primary siRNA screen for histone modulators. MDA-MB-231 cells were transfected with the indicated siRNAs and processed as in (**a**). Three replicate plates ($N = 3$) with identical design were prepared, on which each siCtrl and siKIF11 were transfected in 4 wells, and all other siRNAs were transfected in one well. Quantifications are based on at least 1123 cells per well (excluding siKIF11). Histone H3 signal per DAPI signal was quantified separately for G1, S and G2/M phases, and the well-wise H3 signal was calculated as mean of these 3 values. The bars represent the mean across replicate wells. Significance against the scrambled contol siRNA was analyzed with two-tailed, unpaired Student's $t$-test with Benjamini-Hochberg correction. **e** High-content imaging analysis of histones H3 and H4 in the secondary siRNA screen. MDA-MB-231 cells were transfected with the scrambled control siRNA and processed as in (**a**). Representative microscopy images are shown. **f** Secondary siRNA screen for histone modulators. MDA-MB-231 cells were transfected with the indicated siRNAs and processed as in (**a**). Two separate experiments were performed for the validations using either polyclonal anti-H3 antibody or the combination of monoclonal H3 and H4 antibodies. In each experiment, cells were transfected with siRNAs in 9 (siCtrl), 3 (siKIF11) or 4 (all other siRNAs) replicate wells. Quantifications are based on at least 1123 and 1174 cells per well for the pAb H3 and the mAb H3/H4 screens, respectively (excluding siKIF11). Histone H3 and H4 signals per DAPI signal were quantified separately for G1, S and G2/M phases, and the mean of these 3 values is represented. Significance against the scrambled contol siRNA was analyzed with two-tailed, unpaired Student's $t$-test with Benjamini-Hochberg correction. **g** Western blot validation of histone modulators. MDA-MB-231 cells were transfected with the indicated siRNAs and protein lysates were produced after 3 days. Protein levels were analyzed by Western blot. The barplot represents histone levels normalized to DHX9. Significance against the scrambled contol siRNA was analyzed with two-tailed, unpaired Student's $t$-test with Benjamini-Hochberg correction ($N = 4$ independent siRNA transfections). **h–j** Western blot and qPCR validation of candidate siRNAs. MDA-MB-231 cells were transfected with the indicated siRNAs. Protein lysates and RNA extracts were prepared after 3 days. Protein levels were analyzed by Western blot (**h, i**). The asterisk indicates that the apparent molecular weigth of DCAF6 was higher than expected, requiring an additional assay to control for DCAF6 depletion. The DCAF6 mRNA level was analyzed by qPCR using GAPDH mRNA as normalization control (**j**). Experiments were performed with three technical replicates. Data are represented as mean ± standard deviation. siCtrl is a scrambled siRNA. The scale bars correspond to 40 µm.

inhibit the expression of nuclear encoded mitochondrial genes by interfering with chromatin remodeling[53]. Since CHCHD4 knock-down impedes mitochondrial respiration, our data raise the possibility that respiration inhibitors in general may down-regulate histone levels. Given the currently explored strategies to specifically kill cancer cells by blocking mitochondrial respiration together with dietary restriction[54,55], it would be interesting to investigate a potential role of histone density modulation on mitochondrial functionality and metabolic adaptation.

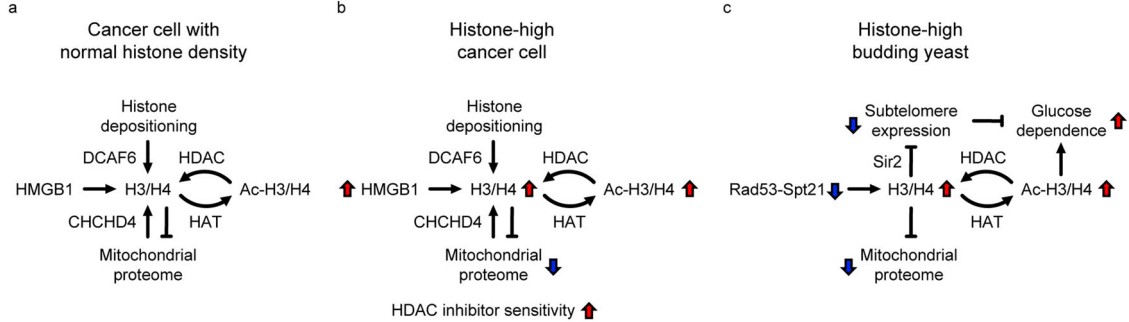

**Fig. 7 Model of conserved responses to aberrant histone density. a, b** Modulators of histone density and cellular processes influenced by histone density in cancer cells. High histone density is associated with elevated histone H3 acetylation, increased HMGB1 expression and repression of mitochondrial proteins. The histone modulators CHCHD4 and DCAF6 may influence histone density during mitochondrial stress responses and DNA replication-linked histone depositioning. **c** Evolutionary conservation of the acetylation reponse and the mitochondrial regulation in budding yeast, Saccharomyces cerevisiae. Aberrant histone accumulation can be caused by a defective Rad53[CHK1/CHK2]-Spt21[NPAT] axis, which controls replication-coupled histone gene expression. Excess histones are hyper-acetylated, silence subtelomeric genes and repress mitochondrial gene expression. Acetylation and subtelomeric silencing together affect central carbon metabolism and cause glucose dependence. Note that the high acetylation state and the repression of mitochondrial protein production are conserved features of histone-high yeast and cancer cells.

Particularly high histone density (>30% increase) is associated with global histone H3 acetylation on lysines 14, 18, 23, 27 (Fig. 7). Intriguingly, excessive histone accumulation in yeast elicits a global histone hyper-acetylation response (Fig. 7)[16]. Such hyperacetylation is likely a protective mechanism to ameliorate excess histone toxicity with substantial side effects on central carbon metabolism that lead to enhanced dependence on glucose metabolism (Fig. 7)[16]. We show that cancer cells with high histone density and acetylation are particularly sensitive to HDAC inhibitors trichostatin-A and panobinostat. Together, these observations imply that excessive histone dosage could sensitize cancer cells to HDAC inhibition by imposing an aberrant global acetylation state. Excessive histone density could sensitizes towards HDAC inhibition in the context of cancer therapy, and it would be interesting to explore histone density as predictive marker for the success of HDAC inhibitor treatments.

We describe a link between histone density and E2F transcription signatures (Fig. 4g). Notably, the mRNA level of *E2F2*, one of the E2F-encoding genes, is elevated in histone-high cells (Supplementary Data 3). Consistent with the absence of correlation between replication-dependent histone mRNAs and proteins, target signatures of the histone gene transcription factors NPAT, Oct-1 and HiNF-1 are not associated with histone density. Hence, E2F, which is not a classical histone gene regulator[8], could indirectly control histone density. Interestingly, high histone density is also associated with resistance to 4 different MEK1/2 inhibitors (Fig. 5e). Resistance to MEK inhibitors is usually linked to the re-activation of mitogen-activated protein kinase (MAPK) pathways, or with the activation of parallel pro-growth signaling pathways, such as PI3K, STAT and Hippo signaling[56]. The E2F transcriptional signature in histone-high cell lines argues for enhanced pro-growth signaling, which activates the G1/S transition by stimulating G1- and S-phase CDK and hence E2F activity[57]. While such enhanced signaling could account for the MEK inhibitor resistance, the mechanistic basis for the activation of growth signaling and its link to histone density need to be further investigated.

DCAF6 interacts with the Cul4A/DDB1 E3 ubiquitin ligase to modulate its activity towards the androgen receptor[58]. Interestingly, Cul4A/DDB1 is crucial for nucleosome assembly by ubiquitylation of histone H3 on lysine 56[59]. Depletion of DCAF6 reduces histone dosage and replication speed, increases S and G2/M phase populations and ultimately reduces cell proliferation (Fig. 6f and Supplementary Fig. 7). These phenotypes are consistent with insufficient histone supply during DNA replication, and we hypothesize that DCAF could modulate H3 by assisting its ubiquitylation during replication-coupled nucleosome assembly.

Correction of histone levels with the nuclear index has one major function: It facilitates an organelle normalization of the almost exclusively nuclear histones, and thereby compensates for variations in nucleus-to-cytoplasm ratio that commonly occur between lineages, cell types and individual cell lines. As a result, nuclear index correction globally reduces co-expression scores while preserving well-characterized co-expressions across histones. In contrast to the normalization with a single nuclear reference protein, nuclear index correction integrates information on various protein complexes and takes into account the actual quantitative relationship between histones and the nuclear index across cell lines. This enhances the robustness against de-regulation of individual proteins and over-correction, respectively. Although DNA content is linked to the amount of DNA-binding proteins, it is important to note that nuclear index-corrected histone densities do not directly reflect histones per DNA. Importantly, we show for cancer cell lines from 3 different lineages that the results obtained from DNA content normalization are consistent with histone densities predicted by nuclear index correction. We recommend a similar validation when working with cell lines of predicted extreme histone density. We strongly emphasize that protein and nuclear DNA measurements should be performed on the same sample because histone and DNA content are affected by cell cycle distribution, which is influenced by multiple experimental conditions. A clean DNA normalization across all CCLE cell lines would require matched measurements of proteomics and nuclear DNA content population means. Future proteomics studies may allow such direct comparisons to identify cases in which nuclear index correction produces outcomes that deviate from DNA normalization.

DNA sequencing and RNA-Seq are the strandard technique to identify mutations, copy number variation and gene expression in large cancer projects[60]. Proteome datasets are becoming available, but cover less patients than sequencing-based methods[33,35,37]. We were able to demonstrate that mRNA levels of replication-dependent histones are not a suitable predictor of their protein levels, emphasising the need of protein quantification when estimating histone de-regulation in cancer. We provide the histone density-associated proteome with a protein ranking by predictive power (Supplementary Data 6–7). Our analysis is

restricted to cancer cell lines, and applicability of nuclear index correction and histone density prediction markers to clinical samples will need to be demonstrated. Our study thus establishes a starting point to address the impact of histone density in cancer.

## Methods

**Processing of datasets**. BioMart annotations for gene conversions were downloaded from the BioMart website on the 5th of October 2020 and manually complemented with annotations from genenames.org. The annotations were used for all conversions. HISTome2[6] annotations were used for the mapping of histone genes to histone protein variants.

CCLE protein expression, mRNA expression by RNA-Seq (DepMap Public 20Q2), mRNA expression by microarray and sample information (DepMap Public 20Q2) files were downloaded from the DepMap portal. Gene and protein IDs were converted to HGNC symbols. Duplicate measurements across proteins or samples were averaged. Genes and proteins with zero expression in all cancer cell lines were exluded. For genome-wide analyses, mRNAs and proteins detected in less than 5 or 150 cell lines were excluded. Data were lineage-centered unless otherwise indicated by centering the log$_2$ expression of each gene or protein to 0 for each lineage.

Clinical Proteomic Tumor Analysis Consortium datasets and sample information files were downloaded from the Clinical Proteomic Tumor Analysis Consortium data portal. TCGA datasets and sample information files were downloaded from the GDC data portal. All Clinical Proteomic Tumor Analysis Consortium datasets and the TCGA colon adenocarcinoma, breast cancer and ovarian cancer files were downloaded in matrix format. TCGA lung squamous cell carcinoma, glioblastoma multiforme, head and neck squamous cell carcinoma and uterine corpus endometrial carcinoma datasets in individual files per patient and merged into expression matrices. Gene and protein IDs were converted to HGNC symbols. Expression levels were averaged per patient. Colon adenocarcinoma proteome data were normalized by variance stabilizing transformation using the DEP R package.

**Integration of mRNA levels for histone variants**. Transcript quantification in the CCLE gene expression dataset uses the software package RSEM[61], which allows the accurate estimation of transcript abundance for highly similar genes, such as genes encoding the same histone variant. We converted expression values to the linear scale by reverting the the log$_2(x + 1)$ transformation used in the CCLE expression dataset, calculated the sum of all linear values for each histone variant and applied the log$_2(x + 1)$ transformation to obtain logarithmic values of histone variant expression ($x_{variant}$):

$$x_{variant} = \log_2(1 + \sum(2^x - 1)) \qquad (1)$$

**Data visualization**. We used the R packages ggplot2 (general plotting), Rcolor-Brewer (color palettes), ggVennDiagram (Venn diagrams), ggraph/igraph (edge bundling), ggpubr (emapplot) and pheatmap (heatmaps with hierarchical clustering) for data visualization.

**Analysis of mRNA read distributions**. Genes were categorized as replication-dependent histone gene, replication-independent histone gene or other gene based on the HISTome2 database.

**Correlation analysis**. Correlations of mRNA vs mRNA, protein vs. protein and protein vs. mRNA were performed across cell lines using the R function cor for pairwise complete observations.

**Gene ontology and gene set enrichment analysis**. We used the DAVID Bioinformatics Resources 6.8 online tool for enrichment of cellular compartment gene ontologies. A custom R script was used for the calculation of the cumulative sum of gene ontology sets and for visualization. We used the R package cluster-Profiler to perform gene set enrichment analysis on the log$_2$ mRNA fold-changes in histone-high vs. -low cell lines. We obtained the C3 collection (regulatory target gene sets, version 7.4) from MSigDB as transcription factor target gene sets. We clustered and visualized enriched sets with the R packge clusterProfiler together with ggpubr.

**PCA dimensionality reduction and model training**. We used the R packages FactoMineR and factoextra for principal component analysis (PCA). We used a principal component correlation plot (circular) for the visualization of top histone correlator functions. For the classification of cell lines as histone-high or -low, we performed a PCA on the significantly different proteome between histone-high and -low cells and established a predictive model for histone dosage state (high vs. low) based on the first two principal components. We visualized the 95% confidence intervals for correct classification in the principal component sample plot. We used the R packages caret for model training and ROCR for calculation of ROC statistics and ROC curve visualization. Model training for the ranking of proteins by histone

density group prediction was implemented with parallel computing using the R packages foreach and doParallel.

**Protein network visualization**. We used STRING version 11.5 for protein network construction and visualized the STRING output with Cytoscape version 3.5.1. Proteins were arranged manually in Cytoscape according to the ten nuclear index protein classes.

**Nuclear index calculation**. For each of the 10 protein classes contributing to the nuclear index, a median expression level (the median of its proteins) was calculated. The nuclear index was then calculated as median of the 10 representative protein class values. The nuclear index is based on lineage-centered data. However, a calculation without lineage centering is possible and was used to analyze the interaction between lineage and nuclear index effects on histone levels.

**Nuclear index correction**. For each protein of interest, a quantile regression (R package quantreg) was performed against the nuclear index over all indicated cell lines. The corrected protein expression was calculated for each cell line by subtracting the predicted expression for the given nuclear index from its measured expression in the CCLE dataset.

**Cell line classification and significance analysis of differentially expressed features**. We classified cell lines as histone-high or -low based on a least 20% increase or decrease of both histones H3.1 and H4, using lineage-centered, nuclear index-corrected protein expression data. We restricted all types of significance analysis to lineages in which both histone-high and -low cell lines were represented (blood, breast, central nervous system, gastric, kidney, liver, lung, lymphocyte, ovary, pancreas, skin, upper aerodigestive). For significance analysis of protein expression, we used the R package maanova (FDR < 0.1, fold-change > 1.3). For significance analysis of mRNA expression and other features (metabolites, histone modifications, drug sensitivity), we applied the R package maanova (FDR < 0.1) to lineage-centered data. We validated that all of the differentially expressed proteins were also statistically significant when applying proteome-wide nuclear index correction (Supplementary Data 2), and that most of differentially expressed proteins were consistently altered when comparing histone-high or -increased low vs. cell lines with normal histone density (Supplementary Fig. 5a).

**Cell culture, media and treatments**. Cells cultured in RPMI 1640 medium (TermoFisher) with 10% fetal bovine serum, 2 mM glutamine (Life Technologies) and penicillin/streptomycin (Life Technologies), in a humidified incubator atmosphere at 37° and 5% CO$_2$. A custom cherry-pick siRNA library in the SMARTpool format was obtained from horizon, with KIF11 siRNA (FE5L003317000005) and non-targeting pool siRNA (FE5D0018101005) as controls. siRNA transfection was done 3 days before cell fixation or lysis. We used OptiMEM (ThermoFisher) and Lipofectamine RNAiMAX (ThermoFisher) reagents for siRNA transfection. For each 96-well, 0.2 μL RNAiMAX in 10 μL OptiMEM were mixed with 10 μL OptiMEM containing 120 nM siRNA. The mix was added to the wells and incubated for 20 min. 100 μL cells in culture medium were then seeded on top at 10–15% confluency. For 6-well transfections the volumes were scaled up 30-fold. For immunocytochemistry, 5-ethynyl-2'-deoxyuridine (EdU, ThermoFisher) was added to a final concentration of 4 μM 1 h before fixation. Cell lines were obtained from ATTC (MDA-MB-157, MDA-MB-453, WM-266-4, A-172, U118MG), NCI (MDA-MB-231, UACC-62), CLS (MDA-MB-436), DSMZ (IPC-298, IGR-1), and ICLC (MEWO). None of the used cell lines is listed as misidentified cell line in the ICLAC register. Cell morphologies were tested visually for all cell lines. Cell line validation was performed by STR profiling (gene print 10 system, Promega). All cell lines were tested negative for mycoplasm contamination by PCR and colorimetric assay (mycoalert detection kit, Lonza). Bright-field images of all used cell lines are provided in Supplementary Fig. 4a.

**Cell lysis and immunoblotting**. Total cell lysates were prepared in lysis buffer (50 mM Tris-HCl pH 8.0, 1 mM MgCl$_2$, 200 mM NaCl, 10% Glycerol, 1% NP-40) with EDTA-free protease inhibitor cocktail (Roche). Protein concentrations were quantified by Bradford assay (Bio-Rad) and equal amounts of protein were boiled with Laemmli buffer. Samples were resolved using Bolt$^{TM}$ 4–12% Bis-Tris Plus precast gels (Invitrogen) with MES buffer (Invitrogen), transferred to a 0.2 μm nitrocellulose membrane for 16 h at 30 V. Proteins on membranes were visualized with Ponceau S solution. Membranes were blocked for 30 min at room temperature with blocking solution (5% non-fat dried milk in 1x TBS with 0.075% Tween-20). Antibodies were diluted in blocking solution. Primary antibodies were incubated overnight at 4 °C and secondary antibodies for 1 h at room temperature. Super-Signal™ West Dura Extended Duration Substrate (ThermoFisher) and a ChemiDoc imaging system (Image Lab v5.0) were used for signal acquisition. ImageJ software (version 1.51d) was used for signal quantification. The preparation of Figure panels was done with ImageJ and GIMP (version 2.8.14), and the cropping of original images is shown in Supplementary Fig. 11. The following antibodies were used for Western blotting: rabbit polyclonal anti-histone H3 (EpiCypher, Cat# 13-0001, 1:5000), rabbit monoclonal anti-histone H4 clone D2X4V (Cell Signaling

Technology, Cat# 13919, 1:4000), rabbit polyclonal anti-DHX9 (Atlas Antibodies, Cat# HPA028050, 1:2000), mouse monoclonal anti-RNA polymerase II (Santa Cruz Biotechnology, Cat# sc-47701, 1:500), rabbit polyclonal anti-CHCHD4 (Novus biologicals, Cat# NBP2-76390, 1:1000), rabbit polyclonal anti-DCAF6 (Novus biologicals, Cat# NB100-56434, 1:1000), rabbit polyclonal anti-SCO1 (Atlas antibodies, Cat# HPA021579, 1:1000), mouse monoclonal anti-ALAS1 (Santa Cruz Biotechnology, Cat# sc-365153, 1:200), mouse monoclonal anti-COX4 (Cell Signaling Technology, Cat# 11967, 1:5000), mouse monoclonal anti-α-Tubulin (Merck, Cat# T5168, 1:1000), goat polyclonal anti-mouse IgG (H + L)-HRP Conjugate (Bio-Rad, Cat# 1706516, 1:20000), goat polyclonal anti-rabbit IgG (H + L)-HRP Conjugate (Bio-Rad, Cat# 1706515, 1:20000).

**cDNA preparation and qPCR analysis**. Three days post siRNA transfection, cells were washed with ice-cold PBS, RNA was prepared for each sample using the RNAeasy Mini kit (Qiagen, Cat# 74104) according to manufacturer instructions. cDNA was prepared for each sample starting from 1 μg of RNA using the High-Capacity cDNA Reverse Transcription Kit (ThermoFisher, Cat# 4368814) according to manufacturer instructions. cDNA samples were treated with 1 μL RNAse H (Promega, Cat# M4281) for 20 min at 37 degrees and stored at −80 °C. Gene expression analysis was performed by the qPCR-Service at Cogentech-Milano. 5 ng of cDNA was amplified in triplicate in a reaction volume of 10 μL containing the following reagents: 5 μL of TaqMan Fast Advanced Master Mix (ThermoFisher), 0.5 μL of TaqMan Gene expression assay 20x for *DCAF6* and *GAPDH* (ThermoFisher). Real-time PCR was carried out on the QS12k (ThermoFisher), using a pre-PCR step of 20 s at 95 °C, followed by 40 cycles of 1 s at 95 °C and 20 s at 60 °C. Samples were amplified with primers and probes for each target, and for all the targets one NTC sample was run. Raw data (Ct) were analyzed with Biogazelle qbase plus software and the fold change was expressed as Calibrated Normalized Relative Quantity.

**High content microscopy analysis, confocal imaging, bright-field imaging and siRNA screens**. For high content microscopy analysis, cells were fixed for 15 min at room temperature in 3% PFA/0.025% glutaraldehyde, and permeabilized with 0.3% Triton X-100 for 10 min. Glutaraldehyde was quenched with 0.1% NaBH$_4$ in PBS for 10 min. The click reaction was performed for 1 h at room temperature in PBS with 2 mM CuSO$_4$, 10 mM sodium ascorbate and 1 μM Alexa Fluor™ 647 Azide (ThermoFisher). Cells were incubated with blocking solution (1% BSA, 5% goat serum, 0.075 Tween-20 in TBS) for 1 h, with the primary antibodies diluted in blocking solution overnight at 4 °C, and with fluorophore-conjugated secondary antibodies for 2 h. Cells were washed three times with TBS-Tween between all incubations. Nuclei were stained with 1 μg/ml 4′,6-diamidino-2-phenylindole (DAPI, ThermoFisher) in PBS. For high-content imaging, images were acquired using a ScanR microscope (Olympus) with a 10x objective, using auto-focus on the DAPI channel and fixed exposure times. Image sets were analyzed with CellProfiler (version 4.1.3)[62] to identify nuclei and measure shape, area and intensities. Normalizations, cell cycle gating, statistics and visualization were performed in a custom R script (provided). At least three independent wells per condition were analyzed, and statistics were calculated over well means. The mean H3 signal per DNA (H3/DAPI) was quantified separately for G1, S and G2/M phases for each screening well to avoid potential bias due to differences in cell cycle distribution, and the mean of these three values was used as representative H3 signal per DNA. We confirmed as measure of transfection and knock-down efficiency that treatment with lethal si*KIF11* reduced nuclei counts to approximately 10% of the scrambled siRNA (siCtrl) (see Supplementary Fig. 10a). Information on the number of analyzed objects is listed in the CellProfiler output files. For confocal imaging, 20,000 cells/well in a 24-well plate were seeded on round slide-glasses pre-coated with fibronectin at 10 μg/ml. 48 h after cell seeding cells were washed once with PBS and fixed with 4% formaldehyde (15 min at room temperature), washed 3 times with PBS (10 min each), permeabilized with 0.5% Triton-X-100 in PBS (5 min at room temperature), incubated with blocking buffer (3% BSA in 0.1% Triton-X-100 PBS) for 1 h, incubated with primary antibody (diluted in blocking buffer) for 1.5 h at room temperature, followed by three PBS washes and incubated with secondary antibodies (1:400 in blocking solution) and phalloidin-FITC (1:50 in blocking solution) for 1 h in the dark at room temperature followed by three PBS washes. DAPI was added in PBS for 5 min at room temperature followed by other 2 washes with PBS. Samples were mounted with Mowiol and stored at 4 °C until image acquisition. Random fields (up to 15) were acquired from each coverslip on an UltraVIEW VoX spinning-disc confocal unit (PerkinElmer), equipped with an Eclipse Ti inverted microscope (Nikon) and a C9100-50 electron-multiplying CCD (charge-coupled device) camera (Hamamatsu), driven by a Volocity software (Improvision; Perkin Elmer). Z-stacks with a step size 0.3 μm were acquired for each field of view for a total Z of 10 μm using a 60X oil objective. Bright-field images were acquired at a EVOS imaging system using a 40x objective. The preparation of Figure panels was done with ImageJ and GIMP, and the cropping of original images is shown in Supplementary Fig. 12. The following antibodies were used for immunocytochemistry: rabbit polyclonal anti-histone H3 (Abcam, Cat# ab1791, 1:200), mouse monoclonal anti-histone H3 clone 1B1B2 (Cell Signaling Technology, Cat# 14269, 1:200), rabbit monoclonal anti-histone H4 clone D2X4V (Cell Signaling Technology, Cat# 13919, 1:200), mouse monoclonal anti-mitochondria (Abcam, Cat# ab92824, 1:500), donkey polyclonal anti-mouse

AlexaFluor-Cy3 (Jackson ImmunoResearch, Cat# AB_2340813, 1:400), donkey polyclonal anti-rabbit AlexaFluor-488 (Jackson ImmunoResearch, Cat# AB_2313584, 1:400).

**Flow cytometry analysis of DNA content**. Cells were trypsinized and counted, 10$^6$ cells were pelleted at 1500 rpm for 10 min, washed once in 1 ml PBS (4 °C) and centrifuged at 3000 rpm for 5 min. Cell pellets were resuspended in 250 μL PBS (4 °C) by pipetting and fixed by adding 750 μL pure ethanol (−20 °C) dropwise while vortexing. Cells were left in fixative for at least one hour on ice. Cells were then washed once in 1 mL PBS with 1% BSA (4 °C). Finally, pellets were resuspended in 1 mL DAPI dilactate (Merck Cat# D9564) at 2 μg/ml in PBS (4 °C) and stained cells were kept overnight at 4 °C until flow cytometry analysis. Volumes were rescaled in order to keep the same cell density in samples with lower number of cells. for 1 h at room temperature. Samples were acquired with Attune NxT (ThermoFisher) with fixed settings for all cell lines and analyzed with FlowJo 10.8.1. Unstained samples were included for all cell lines to confirm that the background intensity was below 1% of the staining intensity.

**Human research participants**. There was no involvement of human participants in this study.

**Statistics and reproducibility**. Significances for comparisons of multiple groups of normally distributed data were calculated with one-way ANOVA with post hoc Tukey HSD test. Significances for pairwise comparison of normally distributed data were calculated with Student's t test (two-sided, unpaired). Benjamini-Hochberg correction was applied to p values for multiple comparisons. Descriptive statistics on histone expression in cancer cell lines and cancer patients were applied to all available samples without exclusion (Fig. 1a–d and Supplementary Fig. 1a–d). Nuclear index calculation was based on cell lines covered by CCLE proteomics and RNA-Seq datasets (N = 372) (Fig. 2a–h and Supplementary Fig. 2). Nuclear index corrections and cell line classification by histone density were applied to the cell lines covered by the CCLE proteomics dataset (N = 373) (Fig. 3a–f and Supplementary Fig. 3a, b). Histone density validation (Fig. 3g, h) and DNA content analysis (Supplementary Fig. 4c–e) were performed with parallel experimental replicate cultures (N = 3), and statistics were calculated separately by lineage and quantified histone. Mitochondrial protein expression validation was performed in 5 histone-high and 5 histone-low cell lines (Fig. 4d, e). Cell line groups by histone density contain 46 (histone-high), 31 (histone-low), and 296 (histone-medium) cell lines (Supplementary Fig. 3c, d). Cell line groups by histone density filtered for lineages that contain at least one histone-high and at least one histone-low cell line contain 37 (histone-high), 31 (histone-low), and 209 (histone-medium) cell lines These groups were used for most statistical analysis between histone density groups (Figs. 4a–c, f, g, 5a, b and Supplementary Figs. 5a, b, 9a–f). A group of high confidence histone-high cell lines (Fig. 5c, N = 10) was used in multiple omics comparisons vs. a pool of histone-low and -medium cell lines (N = 240) (Fig. 5d–f, Supplementary Fig. 8). Representative imaging Figure panels without formal experimental repetition (Supplementary Figs. 4a, 6, and 7) or from high-throughput screening (Fig. 6b, c, e) are shown as illustrations of cell identity and morphology. The primary high-content imaging screen was performed in 3 replicate wells split to 3 plates, with 4 control siRNAs per plate (Fig. 6d and Supplementary Fig. 10a–c). The secondary high-content imaging screens were performed in 4 replicate wells on the same plate, with 9 control siRNAs (Fig. 6f and Supplementary Fig. 10d–g). siRNA valdation experiments were performed with 4 experimental replicates for histone regulation (Fig. 6g and Supplementary Fig. 10h), and in three technical replicates for knock-down quantification (Fig. 6h–j). No data exclusions were performed unless specifically stated. To assure the high confidence experimental replication of histone regulators, high content analysis data were obtained with 3 antibodies in total, and relevant hits were independently validated by Western blot analysis. Sample randomization and blinding were not performed. However, the study is largely based on published data and a high-content imaging approach, which avoids data acquisition bias. The following antibody validations were performed: rabbit polyclonal anti-histone H3 (EpiCypher, Cat# 13-0001): validated by our lab[16], expected molecular weight in Western blot, expected nuclear signal and correlation with DNA content in immunocytochemistry; rabbit polyclonal anti-histone H3 (Abcam, Cat# ab1791): Validated in manufacturer's website, expected molecular weight in Western blot, expected nuclear signal and correlation with DNA content in immunocytochemistry; mouse monoclonal anti-histone H3 clone 1B1B2 (Cell Signaling Technology, Cat# 14269): Validated in manufacturer's website, expected molecular weight in Western blot, expected nuclear signal and correlation with DNA content in immunocytochemistry; rabbit monoclonal anti-histone H4 clone D2X4V (Cell Signaling Technology, Cat# 13919): Validated in manufacturer's website, expected molecular weight in Western blot, expected nuclear signal and correlation with DNA content in immunocytochemistry; rabbit polyclonal anti-DHX9 (Atlas Antibodies, Cat# HPA028050): Validated in manufacturer's website, expected molecular weight in Western blot; mouse monoclonal anti-RNA polymerase II (Santa Cruz Biotechnology, Cat# sc-47701): Validated in manufacturer's website, expected molecular weight in Western blot; rabbit polyclonal anti-CHCHD4 (Novus biologicals, Cat# NBP2-76390, 1:1000): Validated in manufacturer's website, expected molecular weight in Western

blot, consistent with siRNA knock-down; rabbit polyclonal anti-DCAF6 (Novus biologicals, Cat# NB100-56434, 1:1000): Apparent molecular weight (130 kD) different higher than expected; validated by comparison with literature showing consistent appearance in Western blot at ~130 kD with diverse antibodies detecting DCAF6[63–65], consistent with siRNA knock-down which was confirmed by qPCR; rabbit polyclonal anti-SCO1 (Atlas antibodies, Cat# HPA021579, 1:1000): Validated in manufacturer's website, expected molecular weight in Western blot; mouse monoclonal anti-COX4 (Cell Signaling Technology, Cat# 11967, 1:5000)): Validated in manufacturer's website, expected molecular weight in Western blot; mouse monoclonal anti-α-Tubulin (Merck, Cat# T5168, 1:1000): Validated in manufacturer's website, expected molecular weight in Western blot; mouse monoclonal anti-mitochondria (Abcam, Cat# ab92824, 1:500): Validated in manufacturer's website, typical mitochondrial morphology observed in confocal microscopy in 10 cell lines

**Reporting summary**. Further information on research design is available in the Nature Research Reporting Summary linked to this article.

## Data availability

Source data and uncropped blots are available at Mendeley Data (https://doi.org/10.17632/68pd82kgpg.1)[66]. Uncropped blot image files are located in the Mendeley Data project raw data folder, within the subfolders named after the corresponding Figure panels. High-content imaging raw data are available at figshare (https://doi.org/10.6084/m9.figshare.20412639.v1)[67]. All other data are available from the corresponding author on reasonable request.

## Code availability

The projects were managed in Rstudio version 1.0.153, run in a Windows 7 (64 bit) operative system. The code was executed with R version 4.0.2. Code, data files and a user manual are available at Mendeley Data (https://doi.org/10.17632/68pd82kgpg.1)[66]. The following R package versions were used: data.table 1.13.6, dplyr 1.0.2, ggplot2 3.3.3, RcolorBrewer 1.1–2, ggVennDiagram 1.1.0, ggforce 0.3.2, ggraph 2.0.3, igraph 1.2.6, qvalue 2.20.0, maanova 1.58.0, stringr 1.4.0, tidyr 1.1.3, ggpubr 0.4.0, pheatmap 1.0.12, clusterProfiler 3.16.1, FactoMineR 2.4, factoextra 1.0.7, caret 6.0–86, ROCR 1.0–11, foreach 1.5.1, doParallel 1.0.16, quantreg 5.75, extrafontdb 1.0, extrafont 0.17. The following tool and software versions were used: DAVID Bioinformatics Resources 6.8, STRING 11.5 and Cytoscape 3.5.1.

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

## Acknowledgements

We thank all members of the Foiani lab for helpful discussions. We thank the IFOM cell culture facility and Cogentech microscopy unit for experimental support. We thank the reviewers for constructive input. C.B. was supported by fellowships from Associazione Italiana per la Ricerca sul Cancro (AIRC) Fellowship i-Care (Marie Curie co-funded by the European Union, ID 16173) and from Fondazione Umberto Veronesi (Post-doctoral Fellowships 2021 - ref. 4007). This work was supported by grants from Fondazione AIRC under IG 2018 (M.F., ID 21416).

## Author contributions

Conceptualization: C.B. and M.F.; software: C.B.; formal analysis: C.B.; investigation: C.B. and G.B.; data curation: C.B.; writing-original draft: C.B.; writing-review & editing: G.B. and M.F.; visualization: C.B. and G.B.; supervision: M.F.; project administration: C.B. and M.F.; funding acquisition: C.B. and M.F.

## Competing interests

The authors declare no competing interests.
