## [Peer Review File · Communications Biology]

Reviewers' comments:

Reviewer #1 (Remarks to the Author):

Bruhn C et al estimated the link of cancer cell histone density to global acetylation, mitochondrial state and drug resistance in various data of cancer cell lines. They first examined the mRNA and protein correlation of histone level and revealed that histone mRNA measurements do not reflect histone protein levels in CCLE dataset including 1300 RNA-seq data and 375 proteomics data of cell lines. Next, they searched for nuclear reference proteins to elucidate the histone density, and found the co-expressed proteins with H4 and H3.1 as a nuclear reference for histone expression. From these results, the authors calculated representative nuclear protein expression values, and named them as nuclear index (NI). Moreover, the authors revealed the combination of NI correction and lineage centering refined the evaluation of histone density. With these methods, the authors classified cancer cell lines to histone-high and histone-low, and found that histone density associated with specific proteomic signatures, linking to the expression of chromatin components, mitochondrial proteins, and histone regulators. Histone-high cell lines were associated with acetylation of H3 and drug sensitivity. Finally, the authors identified CHCHD4 and DCAF6 as bona fide histone density modulators with si-RNA-based screening. Totally, the experimental design is straightforward, and the results are convincing.

Major comments

- 1) The authors demonstrated that CHCHD4, a mitochondrial protein, modulates histone density. Is there any relation of mitochondrial protein expression level to histone density? Although the bioinformatic analysis was done, the immunocytochemical examination of mitochondrial proteins in 10 histone-high cell lines would strengthen the conclusion.
- 2) Is there any cytological difference in histone-high and histone-low cell lines? The cytological images of these cell lines would be necessary.
- 3) Although the novel strategy for evaluating histone density is meaningful, the current results were obtained from cell lines but not from tissues. The limitation of the current conclusion (that obtained from in vitro cell lines) would be described.

Minor comment

In line 4 of the third paragraph of introduction, "in in" is changed to "in".

Reviewer #2 (Remarks to the Author):

The manuscript from Bruhn and colleagues uses a variety of data from the Cancer Cell Line Encyclopedia to try to identify cancer cell lines that have altered density of histone proteins. Identifying cell lines with increased or decreased levels of core histones may provide information on regulators of histone abundance or provide insight into the involvement of histone levels in regulating cellular processes. The primary issue with this manuscript is the methodology used to determine whether a cell line is "histone high" or "histone low". The authors use a nuclear index to compare different cell lines that takes into account the abundance of a variety of nuclear proteins. However, this method misses what is likely to be the most important factor driving histone abundance in cells. As the authors state, cancer cells have very fluid genomes and are often aneuploid. While the authors show that the CNV of the histone genes do not correlate with the abundance of histone proteins, the key variable in comparing cell lines is the over DNA content of the cells. Variations in the abundance of one or two chromosomes, would require a significant change in the amount of histones necessary to assemble the genome into chromatin. The authors should use total genome abundance to normalize the amount of histones present in the cell lines. Other issues include the lack of controls in Figure 6 showing that the siRNAs are actually decreasing the levels of the target proteins. In addition, there are many points in the manuscript where the authors significantly overstate the conclusions that can be drawn from the data.

Response to Reviewers

Reviewer #1 (Remarks to the Author):

Answers are provided in red

Sections cited from the new manuscript are with yellow background. Sections in the manuscript file related to the new content are also highlighted in yellow.

Bruhn C et al estimated the link of cancer cell histone density to global acetylation, mitochondrial state and drug resistance in various data of cancer cell lines. They first examined the mRNA and protein correlation of histone level and revealed that histone mRNA measurements do not reflect histone protein levels in CCLE dataset including 1300 RNA-seq data and 375 proteomics data of cell lines. Next, they searched for nuclear reference proteins to elucidate the histone density, and found the co-expressed proteins with H4 and H3.1 as a nuclear reference for histone expression. From these results, the authors calculated representative nuclear protein expression values, and named them as nuclear index (NI). Moreover, the authors revealed the combination of NI correction and lineage centering refined the evaluation of histone density. With these methods, the authors classified cancer cell lines to histone-high and histone-low, and found that histone density associated with specific proteomic signatures, linking to the expression of chromatin components, mitochondrial proteins, and histone regulators. Histone-high cell lines were associated with acetylation of H3 and drug sensitivity. Finally, the authors identified CHCHD4 and DCAF6 as bona fide histone density modulators with si-RNA-based screening. Totally, the experimental design is straightforward, and the results are convincing.

We want to thank this reviewer for his overall very positive view on our manuscript.

Major comments

1) The authors demonstrated that CHCHD4, a mitochondrial protein, modulates histone density. Is there any relation of mitochondrial protein expression level to histone density? Although the bioinformatic analysis was done, the immunocytochemical examination of mitochondrial proteins in 10 histone-high cell lines would strengthen the conclusion.

We have now included additional experiments in the manuscript to further explore the connection between histone density and mitochondrial proteins. We performed our analysis on the 10 cell lines previously validated in the manuscript as histone high/low which are available in our campus cell collection. First, we selected mitochondrial proteins for which our analysis predicted different levels in histone-high and -low cell lines (SCO1, ALAS1), and we validated their expression differences by Western blot analysis (Figure 4d, e). Our initial proteomic analysis suggested that histone density affects a subset of mitochondrial proteins (rather than the total mitochondrial proteome). To confirm this, we tested the expression of COX4 which was not predicted to be linked to histone dosage (Figure 4d, e), and, as expected, COX4 protein levels did not correlate with histone dosage. We have included sections in the Results and Discussion to make a clear distinction between specifically altered mitochondrial proteins vs. mitochondrial protein content in general. We further corroborated this argument by performing confocal microscopy analysis of mitochondria in histone-high and -low cell lines (Supplementary Figures 6 and 7). Naturally, mitochondrial content and structure are highly variable between cancer cell

lines; however, based on our analysis we conclude that there are no obvious systematic differences in mitochondrial distribution, amount or networks related to histone dosage. We have performed these analyses in the established histone-high and -low cell lines of 3 different lineages (skin, breast, CNS) introduced in Figure 3g. We think that the obtained results are clear and contribute to the improvement of the manuscript, and we want to thank the reviewer for raising this point.

Relevant new sections:

- Results page 7:

Western blot analysis of two mitochondrial proteins, SCO1 and ALAS1, confirmed their predicted differential expression (Figs. 4d, e). In contrast, the mitochondrial protein COX4, which was not predicted to be differentially expressed, did not correlate with histone levels (Figs. 4d, e). This implies that histone dosage is linked to specific mitochondrial proteome signatures but not to a global alteration of mitochondrial proteins. Consistently, confocal imaging of mitochondria did not reveal obvious systematic differences in mitochondria content or network structure in histone-high and -low cells (Supplementary Figs. 6, 7).

Figure 4

- Discussion page 10:

While altered histone density is not associated with obvious changes in global mitochondrial network structure or content (Supplementary Figs. 6, 7), it would be interesting to investigate a potential impact on the various metabolic mitochondrial functions in the future.

- See new Supplementary Figures 6 and 7:

Supplementary Figure 6

Supplementary Figure 7 (magnification of mitochondria panels in Supplementary Figure 6)

2) Is there any cytological difference in histone-high and histone-low cell lines? The cytological images of these cell lines would be necessary.

We have now included bright-field and confocal microscopy images of actin cytoskeleton, nucleus and mitochondria (Supplementary Figures 4a, 6 and 7). We have further added a flow cytometry analysis of cell cycle distributions and DNA content, which we used for an independent normalization of histone data (Supplementary Figure 4c-e). Overall, these additional data show that our histone quantifications are not related to systematic alterations in cell morphology or DNA content. In a single cell line, MDA-MB-453, high histone density was related to an overall small cell size. However, the histone-to-DNA normalization helped us to validate the high histone density in spite of the specific cell morphology. We thank the reviewer for the suggestion, which significantly increased the confidence in our experimental models.

Relevant new sections:

- Results page 6: reference to morphological images at first mentioning of cell lines:

As validation, we quantified histone expression in histone-high and -low cell lines of three lineages (breast, skin, CNS) (Supplementary Fig. 4a) by Western blotting, using two NI proteins, DHX9 and RNA polymerase II subunit RPB1 (POLR2A), as reference.

Supplementary Figure 4

- Results page 7: analysis of DNA content:

Since NI calculation is not based on DNA content data, NI-corrected histone densities do not directly reflect histone:DNA ratios, which are relevant for most biological effects of histones. To address how NI-corrected histone densities relate to histone:DNA ratios, we performed quantitative flow cytometry analysis of DNA content in the histone-high and -low cell lines using the DNA binding dye 4',6-diamidino-2-phenylindole (DAPI) (Supplementary Fig. 4c). There were no systematic differences of DNA content between histone-high and -low cell lines, and the NI-corrected histone density classification was in good agreement with histone-DNA ratio across cell lines (Supplementary Fig. 4d, e).

Supplementary Figure 4

- See also new Supplementary Figures 6 and 7 inserted above

3) Although the novel strategy for evaluating histone density is meaningful, the current results were obtained from cell lines but not from tissues. The limitation of the current conclusion (that obtained from in vitro cell lines) would be described.

We have now added the following section to the Discussion to emphasize that our study is restricted to cancer cell lines and requires validation in clinical samples:

- Discussion page 12:

Our analysis is restricted to cancer cell lines, and applicability of NI correction and histone density prediction markers to clinical samples will need to be demonstrated. Our study thus establishes a starting point to address the impact of histone density in cancer.

We believe that this statement will avoid over-interpretation of our results.

Minor comment

In line 4 of the third paragraph of introduction, “in in” is changed to “in”.

Thank you for the correction!

Reviewer #2 (Remarks to the Author):

Answers are provided in red

Sections cited from the new manuscript are with yellow background. Sections in the manuscript file related to the new content are also highlighted in yellow.

The manuscript from Bruhn and colleagues uses a variety of data from the Cancer Cell Line Encyclopedia to try to identify cancer cell lines that have altered density of histone proteins. Identifying cell lines with increased or decreased levels of core histones may provide information on regulators of histone abundance or provide insight into the involvement of histone levels in regulating cellular processes. The primary issue with this manuscript is the methodology used to determine whether a cell line is “histone high” or “histone low”. The authors use a nuclear index to compare different cell lines that takes into account the abundance of a variety of nuclear proteins. However, this method misses what is likely to be the most important factor driving histone abundance in cells. As the authors state, cancer cells have very fluid genomes and are often aneuploid. While the authors show that the CNV of the histone genes do not correlate with the abundance of histone proteins, the key variable in comparing cell lines is the over DNA content of the cells. Variations in the abundance of one or two chromosomes, would require a significant change in the amount of histones necessary to assemble the genome into chromatin. The authors should use total genome abundance to normalize the amount of histones present in the cell lines.

We thank this reviewer for raising this point. The nuclear index, which we use for histone normalization, is a measurement that represents the average nuclear proteome of each analyzed sample as an internal control. As such, it minimizes organelle bias (nucleus/cytoplasm ratio), and accounts for variance in nuclear size that is influenced by various factors such as DNA content and cell cycle stage. However, the nuclear index is not a direct measurement of DNA content. Therefore, we agree that it is important to investigate how the histone densities obtained by NI correction relate to the histone-to-DNA ratio. To address this issue experimentally, we have performed quantitative analysis of single cell DNA content by flow cytometry (Supplementary Figure 4c). Based on these data we estimated the mean population DNA content (Supplementary Figure 4d) and used this measure to normalize our experimentally determined histone levels (Supplementary Figure 4e). Importantly, the obtained histone-to-DNA ratio was in excellent agreement with the histone-high and -low classifications based on the histone densities that we had predicted based on NI correction.

- Results page 7:

Since NI calculation is not based on DNA content data, NI-corrected histone densities do not directly reflect histone:DNA ratios, which are relevant for most biological effects of histones. To address how NI-corrected histone densities relate to histone:DNA ratios, we performed quantitative flow cytometry analysis of DNA content in the histone-high and -low cell lines using the DNA binding dye 4',6-diamidino-2-phenylindole (DAPI) (Supplementary Fig. 4c). There were no systematic differences of DNA content between histone-high and -low cell lines, and the NI-corrected histone density classification was in good agreement with histone-DNA ratio across cell lines (Supplementary Fig. 4d, e).

Supplementary Figure 4

In spite of these supporting data, we are aware that NI correction is not equivalent to DNA normalization. To make this point clear, we included the following paragraph in the Discussion section (page 12):

Although DNA content is linked to the amount of DNA-binding proteins, it is important to note that NI-corrected histone densities do not directly reflect histones per DNA. We show for a range of experimental model cell lines that normalization of experimentally determined histone levels by total population ploidy results in histone-to-DNA ratios that are consistent with our predictions based on NI correction. We recommend such validation when working with cell lines of predicted extreme histone density to assert that the alteration in histone density is accompanied by an alteration in histone-to-DNA ratio. We strongly emphasize that protein and nuclear DNA measurements should be performed on the same sample because histone and DNA content are affected by cell cycle distribution, which is influenced by multiple experimental conditions. A clean DNA normalization across all CCLE cell lines would require matched measurements of proteomics and nuclear DNA content population means. Future proteomics studies may allow such direct comparisons to identify cases in which NI correction produces outcomes that significantly deviate from DNA normalization.

We have considered the use of genomics data to derive an estimate of total DNA content as an alternative normalization reference. We would like to point out the two major limitations of such approach:

- 1) The total DNA content of a representative G1 cell of a given cell line could be used for DNA content normalization. However, the proteomics data are an ensemble measurement of a cell population of an undefined cell cycle distribution. As such, the histone levels cannot be interpreted as the histone levels of a G1 cell but as a population mean. Consequently, cell cycle distributions of all proteomics samples would be required as additional data to consider the differences in cell cycle distribution that depend on cell line and culture conditions, but these data are not available. At a histone density cell line classification threshold of 20% the impact of expected cell cycle shifts can be substantial.
- 2) The larger the cytoplasm of a cell, the smaller is the relative contribution of histones to the total proteome (organelle bias). Using DNA normalization on proteomics data misses this critical point of organelle differences, which nuclear index correction has been developed for. Sequential NI correction and DNA normalization cannot be used to solve this issue because it is very unlikely that NI and DNA content are independent across cell lines.

Overall, we feel that new data and Discussion section help to critically assess the difference of NI correction vs. DNA normalization, while providing necessary recommendations to the reader for experimental validation, and we thank this reviewer for the valuable suggestion.

Other issues include the lack of controls in Figure 6 showing that the siRNAs are actually decreasing the levels of the target proteins.

We apologize for the lack of these controls. We have now included controls for the relevant siRNAs in Figure 6h-j, which show that the target proteins are depleted efficiently. These include Western blot quantification of CHCHD4 and DCAF6. We observe DCAF6 at approximately 130 kD, which is higher than the expected molecular weight (Supplementary Figure 11). Similar observations have been made previously (e.g. PMID 28758620 Figure S4G at ~ 130 kD, PMID 22177699 Figure 1B at slightly above 130 kD in various cell lines, PMID 29608040 Figure 1A at ~ 130 kD, PMID 33773096 Figure 2B, NRIP-GFP at ~ 160 kD, and PMID 21543494 Figure 3B, NRIP-GFP at ~ 160 kD). We have not been able to find molecular marker-supported evidence for NRIP/DCAF6 migration at the predicted size (~100 kD) in any publication, only on antibody product sheets. Although this strongly suggests that the band that we present is indeed DCAF6, we also provide a qPCR quantification of *DCAF6* mRNA.

- Results page 9:

We confirmed that the siRNAs used in the screen and validation experiments reduced expression of CHCHD4 and DCAF6 (Fig. 6h-j).

Figure 6

- Uncropped images are supplied in Supplementary Figure 11

In addition, there are many points in the manuscript where the authors significantly overstate the conclusions that can be drawn from the data.

We revised statements related to the general conclusions throughout the manuscript, in particular such related to applicability and mitochondrial biology. We removed statements of novelty throughout the manuscript and weakened expressions to avoid over-interpretation of correlative evidence. We now clearly emphasize in the Discussion that our data are restricted to cell line systems, to avoid overstatements related to direct *in vivo* applications:

- Discussion page 12:

Our analysis is restricted to cancer cell lines, and applicability of NI correction and histone density prediction markers to clinical samples will need to be demonstrated. Our study thus establishes a starting point to address the impact of histone density in cancer.

We would like to thank this reviewer for all the constructive criticism which was fundamental to improve the quality of our manuscript.

Reviewers' comments:

Reviewer #1 (Remarks to the Author):

The manuscript was adequately revised according to the reviewer's comments, and added several experiments. The limitation of the conclusion was also described. The revised manuscript met the criteria of acceptance.

Reviewer #2 (Remarks to the Author):

The revised manuscript from Bruhn and colleagues provides additional analysis and controls to address the concerns of the reviewers. The most important issue relates to the influence of total DNA content of the cell on histone content. The authors use a nuclear index (NI) composed of a variety of nuclear proteins to normalize the histone content of a variety of cancer cell lines. The original review argued that determining the ratio of histones to total genomic DNA was a more meaningful measurement than the ratio of histones to other nuclear proteins. In essence, it is hard to envision how histones can function outside their normal regulatory roles if the ratio of histones to genomic DNA is not significantly changed. In addressing this issue, the authors performed these measurements. In the data presented in Supplementary Figure 4E, the authors measure histones relative to genomic DNA for the cancer cell lines that they have labeled as "histone high" or "histone low". It is difficult to determine precisely from the graph, but it looks like only 3 or 4 of the cell lines would still be histone high or histone low relative to genomic DNA. Hence, the NI does not seem to be a suitable method of normalizing histone abundance.